# Stress induced gene expression drives transient DNA methylation changes at adjacent repetitive elements

David Secco[1]*, Chuang Wang[2,3], Huixia Shou[2,3], Matthew D Schultz[4], Serge Chiarenza[5], Laurent Nussaume[5], Joseph R Ecker[4], James Whelan[3,6]*, Ryan Lister[1]*

[1]ARC Centre of Excellence in Plant Energy Biology, The University of Western Australia, Perth, Australia; [2]State Key laboratory of Plant Physiology and Biochemistry, College of Life Science, Zhejiang University, Hangzhou, China; [3]Joint Research Laboratory in Genomics and Nutriomics, Zhejiang University, Hangzhou, China; [4]Genomic Analysis Laboratory, The Salk Institute for Biological Studies, La Jolla, United States; [5]UMR 6191 CEA, Centre National de la Recherche Scientifique, Laboratoire de Biologie du Développement des Plantes, Université d'Aix-Marseille, Saint-Paul-lez-Durance, France; [6]Department of Animal, Plant and Soil Science, School of Life Science, ARC Centre of Excellence in Plant Energy Biology, La Trobe University, Bundoora, Australia

**Abstract** Cytosine DNA methylation (mC) is a genome modification that can regulate the expression of coding and non-coding genetic elements. However, little is known about the involvement of mC in response to environmental cues. Using whole genome bisulfite sequencing to assess the spatio-temporal dynamics of mC in rice grown under phosphate starvation and recovery conditions, we identified widespread phosphate starvation-induced changes in mC, preferentially localized in transposable elements (TEs) close to highly induced genes. These changes in mC occurred after changes in nearby gene transcription, were mostly *DCL3a*-independent, and could partially be propagated through mitosis, however no evidence of meiotic transmission was observed. Similar analyses performed in *Arabidopsis* revealed a very limited effect of phosphate starvation on mC, suggesting a species-specific mechanism. Overall, this suggests that TEs in proximity to environmentally induced genes are silenced via hypermethylation, and establishes the temporal hierarchy of transcriptional and epigenomic changes in response to stress.

*For correspondence: david.
secco@uwa.edu.au (DS); J.
Whelan@latrobe.edu.au (JW);
ryan.lister@uwa.edu.au (RL)

**Competing interests:** The authors declare that no competing interests exist.

**Reviewing editor**: Detlef Weigel, Max Planck Institute for Developmental Biology, Germany

## Introduction

Phosphorus (P) is one of the most important macronutrients for all living organisms, being a key component of nucleic acids and membrane phospholipids, as well as being an essential element for energy-mediated metabolic processes. Plants preferentially absorb this nutrient as inorganic phosphate (Pi), a form of P with low availability and mobility in the soil (*Poirier and Bucher, 2002*). As a consequence, Pi is one of the most limiting nutrients for plant growth and development in most agricultural soils. To overcome these issues, application of large quantities of Pi fertilizers has been the primary strategy to maintain crop yields. Yet, this approach is increasingly economically and environmentally unsustainable, with the reserves of Pi rocks greatly diminishing. It is therefore critical to better understand the molecular mechanisms involved in Pi homeostasis in order to generate plants with increased P acquisition and use efficiency, associated with sustained yields that will contribute to

**eLife digest** Phosphate is an important nutrient for all living organisms. This chemical group forms part of the backbone of DNA molecules, and has a crucial role in many chemical reactions that occur inside cells. Plants in particular need a source of phosphate to grow. This is why agricultural fertilizers are rich in phosphate, but unfortunately, the use of fertilizers is not sustainable. Many researchers are now looking for new ways to maintain high crop yields without chemical fertilizers, and understanding how crops are affected in times of shortage will be pivotal to achieving this goal.

DNA contains coded information in the form of genes, which can either be switched on or off. Chemical marks added to the DNA can earmark genes for activation or inactivation, a bit like handwritten annotations in an instruction manual. One example is the addition of a chemical tag called a methyl group to one of the letters of the DNA code—so-called 'cytosine methylation'. However, little is known about how the pattern of these chemical marks on DNA changes in response to changes in the environment.

Secco et al. investigated changes in cytosine methylation in both rice and *Arabidopsis* plants that had been deprived of phosphate. *Arabidopsis*, or thale cress, is a model plant that is often studied by plant biologists because it is small and grows quickly. The experiments showed that when rice plants were not given enough phosphate, the pattern of DNA methylation changed. This was particularly true around certain genes that help the plants survive in these difficult conditions. Notably, in the absence of phosphate, methylation also occurs more often in DNA sequences called transposable elements that sit close to these useful genes, and less often around other genes.

Transposable elements, also known as 'jumping genes' can move within the genome and thus potentially have damaging effects through altering the DNA sequence. However, DNA methylation normally prevents this from happening. Therefore, the extra methylation observed by Secco et al. may be a cautionary measure to inactivate these transposable elements and limit their potential deleterious effects. Further experiments went on to show that these useful genes seem to be switched on before the DNA of these transposable elements is methylated, implying that the extra methylation observed in these transposable elements is a consequence of the activation of these nearby useful genes.

By contrast, similar experiments performed in *Arabidopsis* reveal a very limited change in DNA methylation when the plants are grown under stressful conditions. This might be because *Arabidopsis* has considerably fewer transposable elements than rice. The next challenge will be to explore how significant the environmentally induced silencing of transposable elements is to the stress responses and genome integrity of crop plants.

improve global food security. Plants have developed a wide set of sophisticated responses aimed at acquiring and utilizing Pi efficiently in order to maintain cellular Pi homeostasis even under Pi limiting conditions (−Pi) (*Rouached et al., 2010*; *Chiou and Lin, 2011*; *Peret et al., 2011*). In −Pi, the expression level of genes encoding high affinity Pi transporters (*PTs*) in the roots increases in order to increase Pi uptake, as well as inducing and secreting acid phosphatases and ribonucleases to mobilize organically bound P. In contrast, the levels of P-containing intermediates such as nucleotides, RNA and phospholipids are dramatically reduced under −Pi, with phospholipids being replaced with sulpho- and galactolipids. As a consequence, genes involved in sulpho- and galactolipids synthesis, such as sulpholipid synthases (*SQDs*) and monogalactosyl diacylglycerol synthases (*MGDs*) are highly up-regulated in −Pi conditions (*Misson et al., 2005*; *Secco et al., 2013a*). Members of the SPX-domain containing protein family (e.g., *SPX*, *PHO1* and *NLA*) have also been shown to be key regulators of Pi homeostasis, being involved in Pi transport and signaling (*Wang et al., 2009*; *Rouached et al., 2010*; *Kant et al., 2011*; *Secco et al., 2012*; *Secco et al., 2013a*; *Puga et al., 2014*; *Wang et al., 2014*).

In addition to the complex regulation observed at the transcriptome level, studies have shown that Pi homeostasis is also regulated by several post-transcriptional mechanisms involving non-coding RNAs, such as miR827, miR399 and *IPS1* (*Franco-Zorrilla et al., 2007*; *Chiou and Lin, 2011*) as well as post-translational changes (*Bayle et al., 2011*; *Lin et al., 2013*; *Park et al., 2014*). However, only a limited number of studies have assessed the potential involvement of altered DNA or histone modifications in response to Pi starvation, and stresses in general (*Sahu et al., 2013*). Smith and

colleagues previously reported that in Arabidopsis (*Arabidopsis thaliana*) the histone variant H2A.Z was deposited at a number of Pi starvation-induced (PSI) genes and that a loss H2A.Z resulted with their de-repression (*Smith et al., 2010*). In addition, using low resolution, non-quantitative and locus-specific methods, several studies have shown the potential involvement of altered DNA methylation in response to stresses (*Labra et al., 2002*; *Chinnusamy and Zhu, 2009*; *Wang et al., 2011a, 2011b*; *Karan et al., 2012*; *Chen and Zhou, 2013*; *Sahu et al., 2013*). Deep sequencing technologies now enable whole genome single base resolution analysis of DNA methylation (*Cokus et al., 2008*; *Lister et al., 2008*), thus enabling global assessment of changes in DNA methylation in response to environmental and developmental cues. Indeed, *Dowen et al. (2012)* previously reported that biotic stress could induce dynamic changes in DNA methylation of repetitive sequences or transposons, often coupled to transcriptional changes of neighbouring genes (*Dowen et al., 2012*). In addition, *Zhong et al. (2013)* recently reported that changes in DNA methylation patterns play a role in the process of tomato fruit ripening (*Zhong et al., 2013*).

Cytosine DNA methylation (mC) is involved in a range of important biological processes, including silencing of repetitive sequences and transposable elements (TEs), genomic imprinting, and stable gene silencing. In plants, DNA methylation exists in all sequence contexts (CG, CHG, CHH, where H = A, C or T) through the activity of multiple genetically distinct pathways (*Law and Jacobsen, 2010*; *Matzke and Mosher, 2014*; *Mirouze and Vitte, 2014*). De novo DNA methylation is mediated by the RNA-directed DNA methylation (RdDM) pathways. In the canonical RdDM pathway, transcripts produced from the RNA polymerase IV (Pol IV) are then copied into dsRNAs by the RNA-dependent RNA polymerase (RDR2) before being processed into 24-nucleotides (nt) small interfering RNAs (siRNAs) by DICER-LIKE 3 (DCL3). These newly generated siRNAs are then loaded onto ARGONAUTE 4 (AGO4) before being guided towards the nascent scaffold of RNAs transcribed by the RNA polymerase V (Pol V) through sequence complementarity. Finally, this complex recruits the DNA methyltransferase DOMAINS REARRANGED METHYLTRANSFERASE 2 (DRM2) to perform de novo methylation in all sequence contexts (*Matzke and Mosher, 2014*). Recently, another RdDM pathway, which is independent of Pol IV and DCL3, has been identified and is referred to as RDR6-RdDM (*Nuthikattu et al., 2013*; *Panda and Slotkin, 2013*; *Creasey et al., 2014*; *Matzke and Mosher, 2014*; *Bond and Baulcombe, 2015*). In this pathway, Pol II derived transcripts are copied by RDR6 into dsRNAs before being processed into 21-22-nt siRNAs by DCL2 and DCL4. These siRNAs can then either induce post-transcriptional gene silencing (PTGS) when loaded onto AGO1 or initiate de novo DNA methylation when associated with AGO2, thus ultimately triggering the canonical RdDM pathway. Maintenance of DNA methylation through replication is mediated by METHYLTRANSFERASE 1 (MET1) and CHROMOMETHYLASE 3 (CMT3) methyltransferases in the CG and CHG contexts, respectively, and are thus referred to as symmetrical methylation, while methylation in the non-symmetrical CHH context has to be established de novo after DNA replication and involves the activities of the DOMAINS REARRANGED METHYLTRANSFERASE 1 and 2 (DRM1/DRM2) and CMT2 methyltransferases (*Finnegan et al., 1996*; *Du et al., 2012*; *Zemach et al., 2013*). Within the plant species studied to date, the general methylation state of particular genomic features are relatively conserved, with TEs often highly methylated in all contexts, and CG methylation commonly located in gene bodies (*Feng et al., 2010*; *Zemach et al., 2010*; *Mirouze and Vitte, 2014*). However, large differences in global DNA methylation levels can be observed amongst plant species potentially associated with different TE content in the various plant genomes. Indeed, the TE-rich genome (~40%) of rice has a much higher aggregate level of DNA methylation than the TE-poor (~15%) *Arabidopsis* genome (*Li et al., 2012*; *Ragupathy et al., 2013*; *Mirouze and Vitte, 2014*).

Given the paucity of past studies assessing the impact of abiotic stresses upon the plant DNA methylome and the temporal relationship between DNA methylation and transcriptional changes, we performed a comprehensive spatio-temporal assessment of the impact of limiting a central plant macronutrient, Pi, upon DNA methylation patterns and transcription, in rice (*Oryza sativa*) and *Arabidopsis*. Using whole genome bisulfite sequencing, we identified species-specific, widespread and mitotically heritable changes in DNA methylation in response to Pi starvation that are particularly enriched at stress responsive genes. These changes in DNA methylation occur after changes in nearby gene expression, and are thus likely a consequence of induced transcription of nearby Pi responsive genes, as well as being largely independent of *DCL3a*. Altogether, we demonstrate a species-specific process in which Pi starvation causes highly localized changes in the genomic

DNA methylation patterns in rice, which may act to repress the potentially deleterious activity of TEs located near genes that are highly induced upon stress.

## Results

### Transcriptional responses of rice to Pi starvation and resupply

A comprehensive time-course experiment of Pi-starved plants was undertaken, spanning medium (3 and 7 days), and long-term (21 days up to 52 days) Pi deprivation (−Pi), as well as both short term (1 and 3 days) and long-term (31 days) recovery (*Figure 1A*). The 52 days time point consisting of 21 days starvation +31 days recovery enabled investigation of the effects of long term resupply on Pi starved plants, and coincided with the emergence of the first panicles and grains (*Figure 1A,B*). Pre-germinated rice seedlings were grown for 14 days in Pi sufficient conditions (0.32 mM Pi) before being transferred to either Pi sufficient (0.32 mM Pi) or Pi deficient (0 mM Pi) media for 21 days. After 21 days of Pi deficient treatment, half of the plants were either maintained under Pi deficient conditions or re-supplied with Pi (0.32 mM) for 1, 3 or 31 days. To confirm the effectiveness of the Pi starvation and resupply treatments, physiological and molecular analyses were performed. Growth analyses revealed that 21 days of Pi deprivation resulted in a 2.3 fold decrease in shoot biomass (*Figure 1—figure supplement 1A*), however no significant differences could be observed for the root biomass after 21 days of treatment (*Figure 1—figure supplement 1B*). Consequently, the root-to-shoot biomass ratio, a parameter often used to assess the efficiency of various nutrient stresses, was significantly altered (t-test, $p < 0.05$), with plants grown under Pi deficient conditions for 21 days showing a 2.2 fold increase compared to plants grown under Pi sufficient conditions (*Figure 1—figure supplement 1C*), which is consistent with previous reports (*Reymond et al., 2006*; *Jiang et al., 2007*; *Secco et al., 2013a*). Pi concentration measurements revealed that 3 days of −Pi led to a threefold reduction in root Pi concentration, while 21 days −Pi resulted in a 6 fold decrease in root Pi concentration compared to plants continuously grown under Pi sufficient conditions (*Figure 1C*). Resupplying Pi for 1 day was sufficient to increase the root Pi concentration >4 fold compared to plants grown under Pi deficient conditions, and within 3 days of Pi resupply the root Pi concentration was similar to that of plants continuously grown under Pi sufficient conditions.

To investigate the transcriptional responses to Pi starvation and resupply, RNA sequencing (RNA-seq) was performed on root samples from all time points, using three biological replicates per condition (Figure 1—source data 1, available at Dryad, *Secco et al., 2015*). Hierarchical clustering of the steady state transcript abundance of the 5570 genes identified as significantly differentially expressed (Cuffdiff, FDR < 0.05) in at least one of the time points revealed a gradual increase in the number and fold change of differentially expressed genes upon −Pi, associated with the length of the Pi deprivation (*Figure 1D,E*; Figure 1—source data 1, available at Dryad, *Secco et al., 2015*). Of note, several phosphate starvation-induced (PSI) marker genes, including the *SPX* genes, *MGD2* and *PTs,* were already induced and showed high steady state transcript abundance after only 3 days of Pi deprivation (Figure 1—source data 1, available at Dryad, *Secco et al., 2015*). Surprisingly, 52 days of Pi deprivation was associated with a decrease in the number and extent of significantly differentially abundant transcripts, including most of the PSI marker genes, potentially due to the concurrent occurrence of panicle development and grain filling. Indeed, a previous study aimed at profiling the shoots of rice grown in the field throughout their life cycle identified two major transcriptome changes, occurring just before panicle differentiation and straight after flowering (*Sato et al., 2011*). In addition, the transcription of some of the PSI genes, including *MGD2* and *PHO2*, was reduced before the panicle differentiation, suggesting that the rice plants undergo a major change in Pi homeostasis at the vegetative-reproductive phase transition (*Sato et al., 2011*).

Resupplying Pi for 1 day after 21 days of starvation was sufficient for the transcript abundance of 40% of PSI differentially regulated genes to return to a level that was not significantly different from the +Pi condition at the matched time point (*Figure 1D,E*). Within 3 days of Pi resupply, the internal root Pi content was similar to that of Pi sufficient plants and the transcript abundance of 80% of the 5570 PSI differentially expressed genes had already returned to levels equivalent to Pi sufficient conditions. After 31 days of Pi resupply, the transcript abundance of only 80 PSI genes remained significantly different compared to Pi sufficient conditions. Overall, the physiological and transcriptional changes associated with Pi starvation and resupply confirmed the effectiveness of the Pi treatments, as well as the capacity of the rice plants to rapidly sense and respond to these changing nutrient conditions.

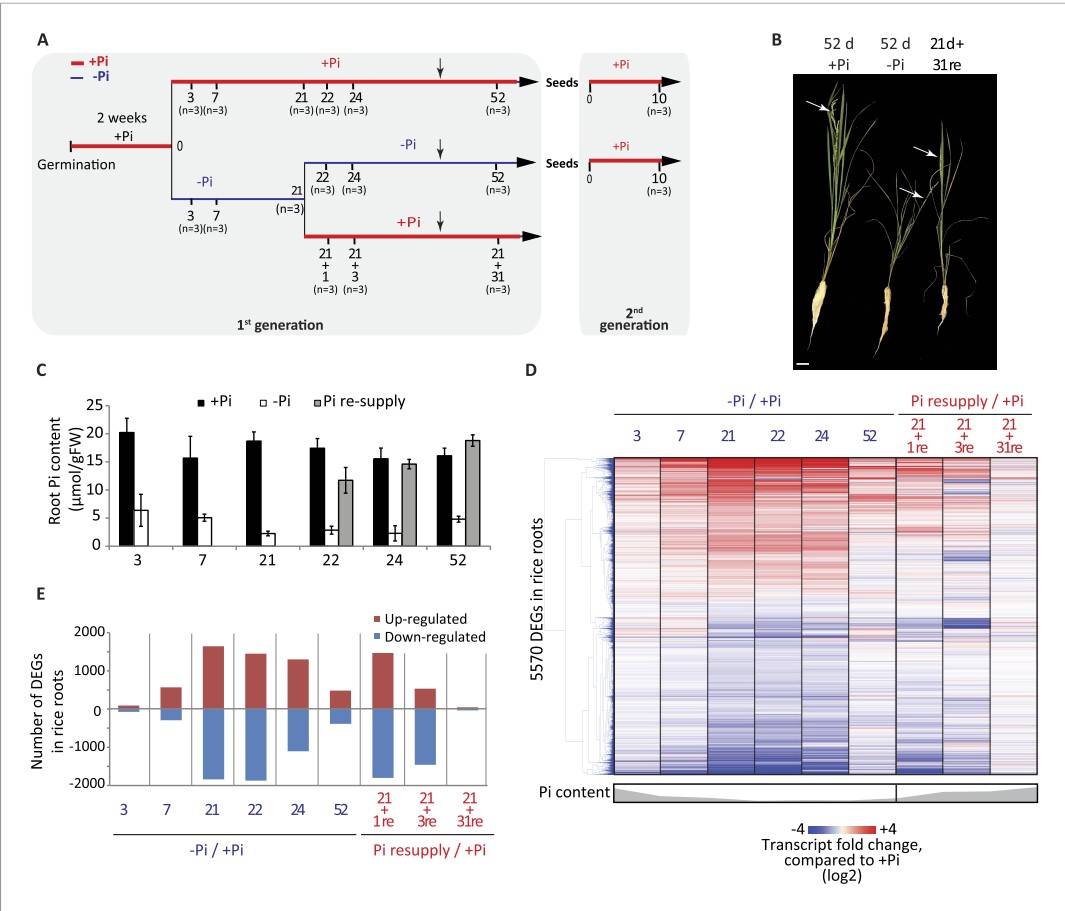

**Figure 1**. Effects of Pi starvation and resupply in rice. (**A**) Schematic representation of the experimental design. Seeds were germinated in water for 2 days and then transferred to a Pi-sufficient hydroponic (0.32 mM) solution for 2 weeks before being either transferred to Pi-deficient media (0 mM) or maintained in Pi-sufficient solution. After 21 days of treatment, half of the Pi-starved plants were resupplied with Pi-sufficient media for up to 31 additional days. Seeds from plants continually grown in + and −Pi were harvested and grown in Pi sufficient conditions for 10 days. Black arrows indicate the time of emergence of panicles. (**B**) Morphological appearance of rice seedlings after 52 days of treatment. White arrows indicate panicles. (**C**) Pi concentration in the roots. (**D**) Hierarchical clustering of significantly (Cuffdiff, FDR < 0.05) differentially expressed genes (DEGs) in response to Pi starvation as determined by mRNA-seq. (**E**) Number of significantly differentially expressed genes in the roots for each time point. Source data for *Figure 1* is available at Dryad (*Secco et al., 2015*).

The following figure supplement is available for figure 1:

**Figure supplement 1**. Responses of rice to Pi starvation.

## Pi starvation induces widespread changes in DNA methylation, enriched at key regulators of Pi homeostasis

To test whether Pi starvation affects genomic DNA methylation in rice, whole genome base resolution profiling of DNA methylation by MethylC-seq was performed on rice roots in triplicate at each time point (*Figure 1A*). Altogether, 45 root single-base resolution high coverage DNA methylomes were generated (77–87% of cytosines covered by at least one read, 82–88% of the genome) (*Supplementary file 1*). In order to identify a set of conserved PSI differentially methylated regions (DMRs) in the roots, the 21, 22 and 24 time points were utilized, resulting in the selection of 9 +Pi samples and 9 −Pi samples. The methylation levels in all sequence contexts (CNN) were then assessed for these samples, and only regions that showed significant differences in methylation levels (FDR < 0.01) in at least 7 of the 9 samples in each of the conditions were considered for further analysis,

resulting in the identification of 175 high confidence root PSI DMRs (Figure 2—source data 1, available at Dryad, *Secco et al., 2015*). Among these 175 PSI DMRs, 84% were hypermethylated in response to Pi starvation (147 hypermethylated, 28 hypomethylated regions, *Figure 2A*). As observed for

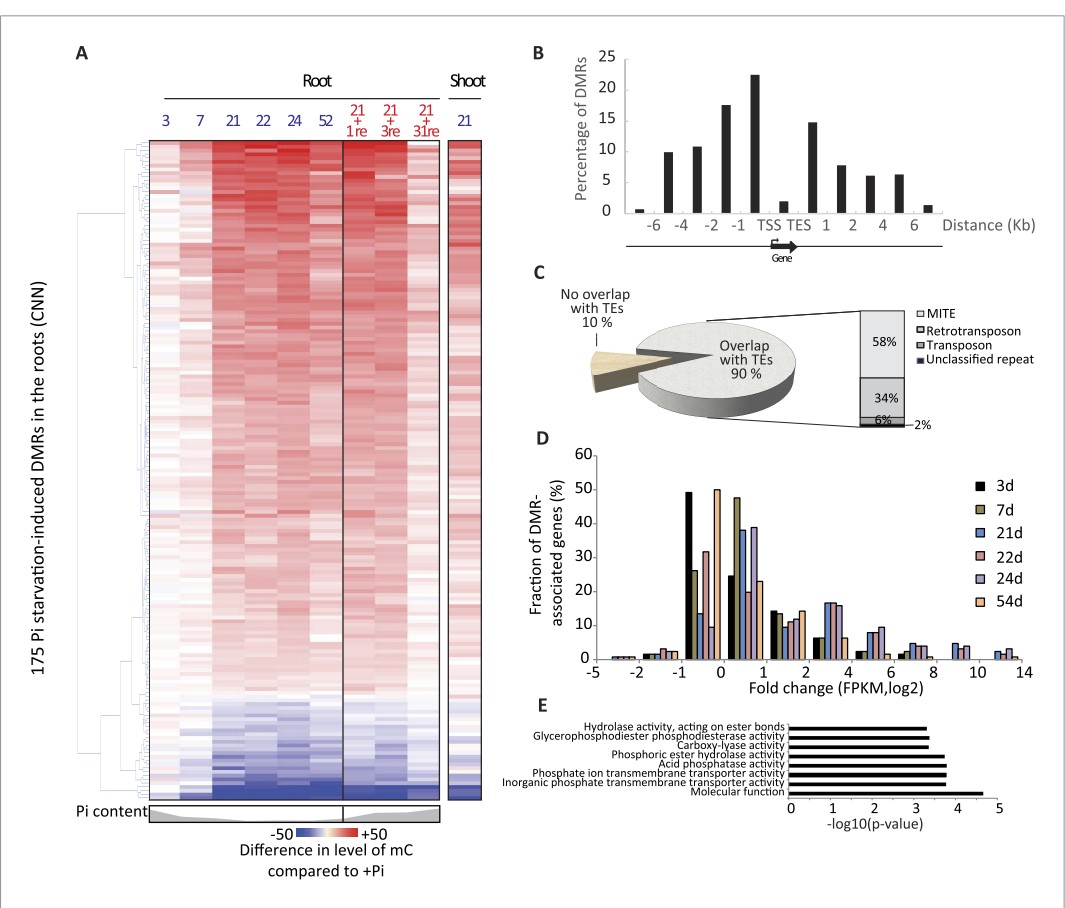

**Figure 2**. Pi starvation triggers widespread changes in DNA methylation in rice roots. (**A**) Hierarchical clustering of the average difference in CNN methylation levels of the 175 PSI DMRs (FDR < 0.01) identified upon long term Pi starvation (21 days, 22 days and 24 days) in the roots in response to Pi starvation. (**B**) Normalised distribution of the distance of the DMRs to the nearest gene. The position of each DMR was calculated with respect to the nearest gene. DMRs were categorized in bins (within gene body, 0–1 kb, 1–2 kb, 2–4 kb, 4–6 kb, and >6 kb from the TSS or TES), and the number of DMRs in each bin was normalised by the total number of regions present in that bin category in the genome. TSS, Transcription Start Site; TES, Transcription End Site. (**C**) Proportion of DMRs overlapping with transposable elements (TEs), and their corresponding classes. (**D**) Distribution of non-redundant DMR-associated gene transcription levels identified by RNA-seq upon Pi deprivation. (**E**) Gene ontology enrichment analysis of non-redundant DMR-associated genes (p-value < 0.05). Source data for *Figure 2* is available at Dryad (*Secco et al., 2015*).

The following figure supplements are available for figure 2:

**Figure supplement 1**. Hierarchical clustering of the 45 root methylomes based on their CNN methylation levels in the 175 root PSI DMRs.

**Figure supplement 2**. Characteristics of the root PSI DMRs.

**Figure supplement 3**. DNA methylation in transposable elements.

**Figure supplement 4**. Examples of PSI DMRs.

**Figure supplement 5**. Hierarchical clustering of the 341 shoot PSI DMRs.

transcript abundance, 3 and 7 days of Pi deprivation resulted in fewer changes in DNA methylation levels, which increased with the duration of the Pi starvation. Indeed, hierarchical clustering of the 45 root samples based on their CNN methylation levels in the 175 PSI DMRs revealed two main clusters, with cluster 1 containing all +Pi samples as well as the 3 and 7 days −Pi samples, and cluster 2 comprising all long-term −Pi time points (≥21 days) in addition to the Pi re-supplied samples (*Figure 2—figure supplement 1*). Furthermore, unlike transcript abundance, where 3 days of Pi resupply was sufficient for the majority of PSI genes to return towards non-stressed level, DNA methylation levels were unaffected by 3 days of Pi recovery. While long-term (31 days) Pi resupply resulted in the PSI DMR methylation level moving towards the Pi sufficient methylation levels (*Figure 2A*), hierarchical clustering analysis revealed that the long-term resupply samples were still more closely related to the long-term Pi starvation samples rather than those of +Pi, as assessed by methylation level at the PSI DMRs in roots (*Figure 2—figure supplement 1*). Thus, nutrient stress-induced differential methylation states can persist for substantial periods of time following cessation of the stress conditions.

The root PSI DMRs had an average size of 205 bp with a mean of 16 differentially methylated cytosines per DMR, and were preferentially localized within the first two kilobases upstream (40%) and first kilobase downstream (15%) of the nearby gene (*Figure 2B*, *Figure 2—figure supplement 2*). Given the role of DNA methylation in repressing TE activity, and the high frequency of TEs in the rice genome, accounting for 40% of the total genome length (*Ragupathy et al., 2013*, *Mirouze and Vitte, 2014*), the position of the DMRs with respect to annotated TEs was assessed. Among the 175 root PSI DMRs, 90% overlapped with TEs, with Miniature Inverted-repeat Transposable Elements (MITEs), the most common class of TEs in the rice genome (69% of all TEs), accounting for 58% of the DMRs overlapping with TEs (*Figure 2C*). To investigate the specificity of the changes in DNA methylation in TEs, DNA methylation levels were assessed in TEs overlapping PSI DMRs as well as in all annotated TEs after 21 days of Pi treatment. Altogether, while TEs overlapping PSI DMRs show significant changes in DNA methylation in response to Pi starvation, the majority of all annotated TEs are unaffected by Pi starvation, suggesting that Pi starvation triggers specific and localized changes in DNA methylation in a subset of all annotated TEs (*Figure 2—figure supplement 3*). In addition, RNA-seq analysis failed to reveal any significantly differentially expressed TEs (Cuffdiff, FDR < 0.05) that overlapped with PSI DMRs, between Pi sufficient and deficient conditions (7 and 21 days), with more than 80% of them not being expressed (Figure 2—source data 2, available at Dryad, *Secco et al., 2015*). Next, each of the 175 PSI DMRs were assigned to the nearest gene, resulting in the association of 126 unique genes with PSI DMRs (Figure 2—source data 1, available at Dryad, *Secco et al., 2015*). Gene transcript abundance analyses revealed that 66 of these DMR-associated genes (52%) were differentially expressed by >2 fold (FDR < 0.05) in response to 21 days −Pi (*Figure 2D*). The steady state transcript abundance of 35% of the 126 genes was >4 fold higher following 21 days Pi starvation. Furthermore, gene ontology analyses revealed a strong enrichment for genes encoding proteins involved in phosphate homeostasis, such as acid phosphatases, phosphate transporters, and factors involved in glycerophosphodiester phospho-diesterase activity (*Figure 2E*, *Figure 2—figure supplement 4*). Of note, while the majority of DMR-associated genes were associated with only one DMR, several genes were associated with multiple DMRs, including key PSI marker genes such as *SPX1*, *SPX2*, *SPX5*, *PT10* and *MGD2*, reinforcing the strong association between differential DNA methylation and transcription in response to −Pi.

To investigate whether Pi starvation could also induce differential methylation in tissues other than the root, which is the primary organ involved in Pi sensing and uptake, high coverage methylomes of rice shoots grown for 21 days under Pi sufficient or deficient conditions were also generated, performing 3 biological replicates per condition (*Supplementary file 1*). Firstly, changes in shoot methylation levels were assessed at the 175 PSI DMRs identified in the roots (*Figure 2A*, Figure 2—source data 1, available at Dryad, *Secco et al., 2015*), revealing similar changes in DNA methylation levels in response to Pi starvation in both roots and shoots, though to a lesser extent in the shoots. Secondly, shoot methylomes under Pi-sufficient and deficient conditions were analysed in order to identify shoot PSI DMRs. Due to the lower number of replicates used to identify shoot PSI DMRs (n = 3) compared to the root PSI DMRs (n = 9), a less stringent FDR cut-off of < 0.05 was selected for further analysis, resulting in the identification of 341 shoot PSI DMRs (FDR < 0.05) (Figure 2—source data 3, available at Dryad, *Secco et al., 2015*). Assigning the 341 shoot PSI DMRs to the nearest genes identified 43 distinct genes that were significantly differentially regulated by Pi starvation in the shoots (*Figure 2—figure supplement 5*). In addition, analysis of the methylation levels in all contexts (CNN) revealed that the 341 shoot PSI DMRs had similar patterns of methylation in both roots and shoots,

suggesting that Pi starvation affects DNA methylation in roots and shoots in a similar manner. Altogether, more than 30 conserved regions showed significant changes in both roots and shoots, and could be associated to PSI marker genes such as *SPX1*, *SPX2* and *MGD2*.

## Pi starvation induced DMRs are mostly hypermethylated in the CHH context

To further examine the potential role of the PSI DMRs, only DMRs that were associated with significant changes in nearby gene transcript abundance following long-term Pi starvation were considered for downstream analysis. Among the 175 root PSI DMRs identified, 100 PSI DMRs were close to genes showing significant changes (FDR < 0.05) in gene transcription upon long term Pi starvation, corresponding to 66 unique genes (*Figure 3A*, Figure 2—source data 1, available at Dryad, *Secco et al., 2015*). Notably, 63 (95%) of these DMR-associated genes were induced by Pi starvation, and included key regulators of Pi homeostasis such as Pi transporters (*PT3*, *PT4*, *PT9*, *PT10*), *SPX* genes (*SPX1*, *SPX2*, *SPX3*, *SPX5*), *MGD2*, *IPS1* and pre-miR827. Only three of the DMR-associated genes were down-regulated by long term Pi starvation. Hierarchical clustering of the differential methylation levels in all contexts for the root PSI DMRs in response to Pi deprivation revealed two distinct clusters, with DMRs in cluster 1 and 2 being hyper- and hypomethylated in response to Pi starvation, respectively (*Figure 3A*, *Figure 3—figure supplement 1*). The first group contained 81 PSI DMRs associated with 61 genes that were overwhelmingly hypermethylated in the CHH context, with a subset displaying CHG hyper-methylation. Furthermore, these hypermethylated DMRs almost exclusively (80 of 81) overlapped with TEs (*Figure 3A*). In contrast, the 19 hypomethylated PSI DMRs from Cluster 2, associated with 13 unique genes, less frequently overlapped with TEs (42% overlap). Notably, most of the known key regulators of Pi homeostasis were present in both clusters, including the *SPX* genes, *IPS1*, pre-miR827 and some *PTs*. Altogether, the majority of PSI DMRs are located in close proximity to genes, gain DNA methylation in response to Pi deprivation, preferentially in the CHH context, and almost exclusively overlap with TEs.

## Changes in gene transcription occur prior to changes in DNA methylation

The detailed time course analysed in this study provides the unique opportunity to decipher the temporal hierarchy between changes in transcription and changes in DNA methylation, which is critical to shed light on the potential causative relationships between the two, and the potential role of the PSI DMRs. The average changes in CNN methylation levels in the PSI DMRs in response to Pi stress were compared to the associated changes in nearby gene steady state transcript abundance (*Figure 3B*, *Figure 3—figure supplement 1*, Figure 2—source data 1, available at Dryad, *Secco et al., 2015*). After 3 and 7 days of Pi starvation, 18% and 55% of the PSI DMR associated genes, respectively, showed significant changes in transcript abundance (Cuffdiff, FDR < 0.05), while only 5% and 9% of the 100 PSI DMRs showed significant changes in DNA methylation compared to +Pi samples (t-test, FDR < 0.05) (*Figure 3B*, *Figure 3—figure supplement 1*, Figure 3—source data 1, available at Dryad, *Secco et al., 2015*). Long term Pi deprivation (≥21 days −Pi) was sufficient to induce significant changes in both gene transcript abundance and DNA methylation levels, with all PSI DMRs being significantly differentially methylated after 24 days of −Pi compared to +Pi (t-test, FDR < 0.05). Furthermore, while resupplying Pi starved plants with Pi for 3 days resulted in 87% of the DMR associated genes differentially expressed upon Pi starvation to return to Pi sufficient-like levels, 94% of PSI DMRs still showed significant differences in DNA methylation levels compared to +Pi conditions (t-test, FDR < 0.05). Finally, while 52 days of Pi starvation resulted in only 22% of PSI DMR associated genes showing significant changes in transcript abundance (Cuffdiff, FDR < 0.05) compared to +Pi, likely a consequence of the floral transition and nutrient reallocation to the grains, the majority of PSI DMRs remained unaffected, with 83% of PSI DMRs still showing significant changes in DNA methylation after 52 days −Pi compared to 52 days +Pi (t-test, FDR < 0.05). Resupplying Pi for 31 days resulted in 11 of the 100 PSI DMRs showing significant persisting changes in DNA methylation (t-test, FDR < 0.05) while no significant changes in gene transcription could be observed compared to +Pi. Taken together, Pi deprivation and Pi resupply appear to first rapidly modulate the transcript levels of genes induced by the stress before subsequently inducing changes in DNA methylation, indicating that the Pi starvation-induced changes in transcription precede, and are potentially causal, for the changes in DNA methylation. This induction of methylation may be involved in repressing the activity of specific TEs close to highly induced genes, via hypermethylation of the TEs in the CHH context.

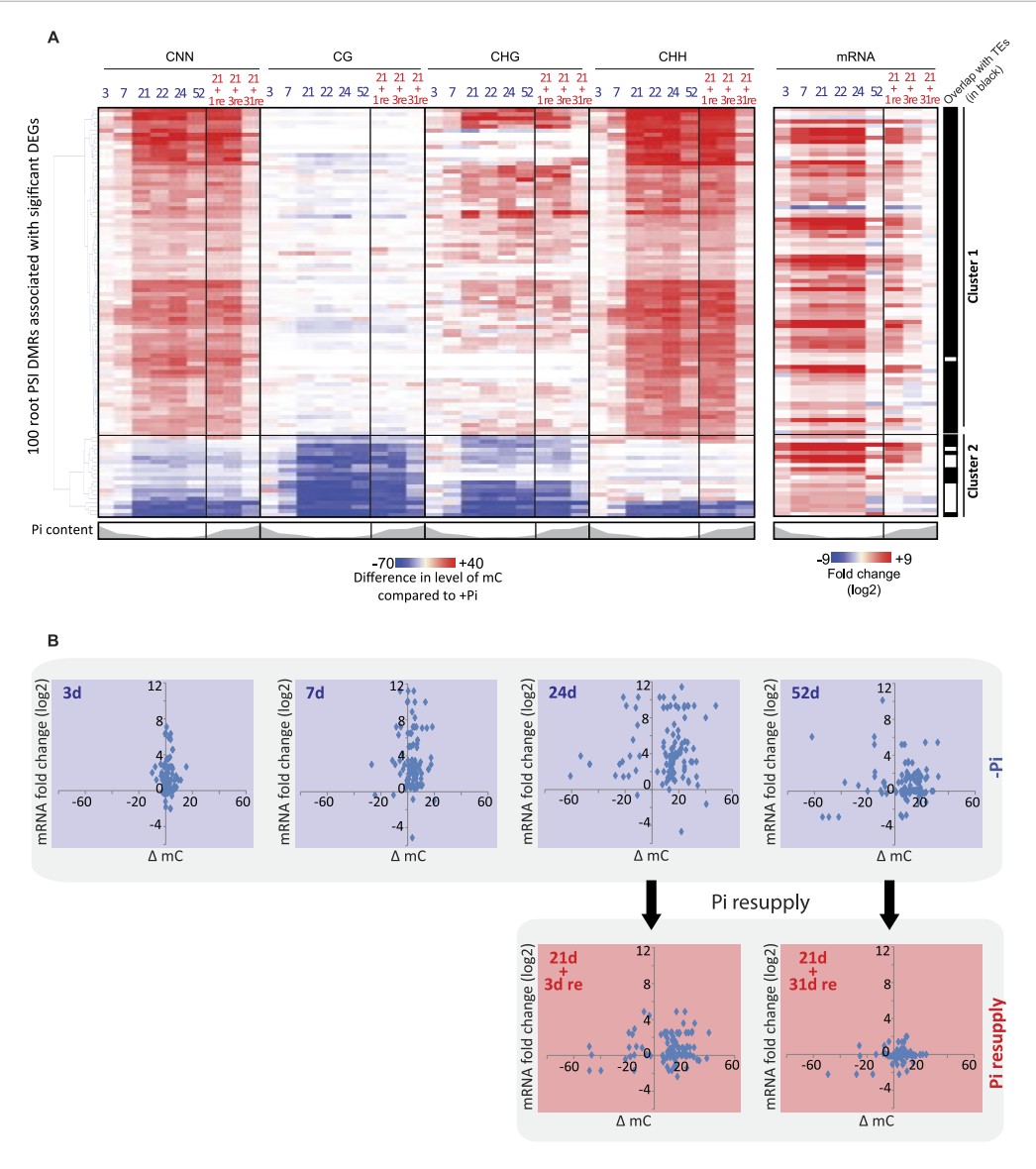

**Figure 3**. Pi starvation-induced DMRs are enriched at key regulators of Pi homeostasis. (**A**) Hierarchical clustering of the differential methylation levels of the 100 root PSI DMRs associated with a significant change in nearby gene expression (DEGs) in response to Pi stress (left panel). Middle panel represents the DMR-associated gene transcript abundance fold change (log2) in response to Pi stresses, while right panel indicates DMRs that overlap with TEs (coloured in black). (**B**) Scatter plots of the changes in DNA methylation (compared to +Pi) (X) against the changes in gene transcript abundance of the nearby associated gene (FPKM, log2) (Y) for each of the 100 PSI DMRs, at various time points. Blue coloured plots represent Pi starvation [(−Pi) − (+Pi)] while red coloured plots represent Pi resupply [(Pi resupply) − (+Pi)]. Source data for *Figure 3* is available at Dryad (*Secco et al., 2015*).

The following figure supplement is available for figure 3:

**Figure supplement 1**. Changes in DNA methylation and gene transcription.

## Phosphate starvation induced DMRs are mainly transient

Due to the observation that differences in DNA methylation in some PSI DMRs may persist despite 31 days of Pi resupply, during which root fresh weight is increased by more than 2.2 fold (*Figure 1—figure supplement 1*), we wanted to assess the extent of potential transmission of changes

in DNA methylation through mitosis. Hierarchical clustering of 45 root samples based on their methylation levels in all sequence contexts (CNN) in the 175 PSI DMRs revealed that the 31 days Pi recovered roots were more closely related to the −Pi samples than the +Pi samples, further indicating that some stress induced differences in DNA methylation can persist despite extended Pi recovery (*Figure 2—figure supplement 1*). However, resupplying Pi for 31 days resulted in 96 of the 175 PSI DMRs showing more than a 2 fold reduction in the differential DNA methylation levels induced by 52 days −Pi, when compared to +Pi, suggesting that the majority of PSI DMRs are returning towards Pi sufficient levels after 31 days of Pi resupply (*Figure 4—figure supplement 1*). Indeed, only 14 of the 175 PSI DMRs (11 hypermethylated, 3 hypomethylated) showed significant differences in DNA methylation level (CNN) between 52 days +Pi compared to 52 days resupply (t-test, FDR < 0.05), while not presenting significant differences between 52 days −Pi vs 52 days resupply (t-test, FDR > 0.05), suggesting that changes in DNA methylation persist in only a limited subset of these PSI DMRs after 31 days of Pi resupply (*Figure 4A*, Figure 4—source data 1, available at Dryad, *Secco et al., 2015*). Of note, despite identifying significant persisting differences in DNA methylation, Pi resupplied samples often showed DNA methylation levels that were intermediate between +Pi and −Pi samples. These persistent differences in methylation levels preferentially occurred in the CHH context (*Figure 4A*) and were associated with 13 unique genes located nearby, including key regulators of Pi homeostasis such as *SPX2* and *MGD2*. Among these 14 persisting PSI DMRs, the single hypomethylated persisting DMR, namely the second DMR associated with *SPX2* (denoted SPX_DMR2), showed the greatest change in DNA methylation level (CNN) in response to Pi stress, decreasing from 50% in +Pi to 1.3% in −Pi, as well as being maintained at a similar low level (1.5%) despite 31 days of Pi resupply (*Figure 4A*). Overall, it appears that only a small proportion of the Pi starvation induced changes in DNA methylation can be sustained despite 31 days of Pi resupply and active cell growth, while the majority of PSI DMRs have DNA methylation levels returning towards +Pi levels.

In order to further investigate the observation that only a limited number of PSI DMRs can be transmitted through mitosis, we analysed a tissue that was only generated post-stress, the panicles from Pi recovered plants (*Figure 1A,B*). Indeed, by the time the panicles appeared (∼40 days after initiation of treatment), the recovering plants had been re-supplied with Pi for ∼20 days, and should thus have a physiological status similar to that of Pi sufficient plants. After 52 days of Pi treatment, panicles were collected from plants continuously grown in +Pi or −Pi, as well as 21 days Pi starved plants resupplied for 31 days, using independent triplicates. Generation of high coverage methylomes enabled the identification of 36 PSI DMRs in the panicles of plants grown for 52 days in +Pi and −Pi (FDR < 0.05). However, no significant differences (t-test, FDR < 0.05) could be observed in the DNA methylation levels of these 36 regions between panicles of plants grown for 52 days in −Pi compared to those resupplied with Pi for 31 days (*Figure 4B*, Figure 4—source data 2, available at Dryad, *Secco et al., 2015*). Yet, using a less stringent FDR cutoff of < 0.1, two of the 36 PSI DMRs identified in the panicles, including SPX2_DMR2, showed persisting differences in DNA methylation despite 31 days of Pi resupply, suggesting that a limited number of PSI DMRs could potentially be transmitted to newly generated tissue that has never experienced the stress.

We next tested whether PSI DMRs could be transmitted from Pi stressed plants to their progeny, and thus potentially act as a transgenerational stress memory mechanism. To do so, individual seeds from five individual plants continuously grown under Pi sufficient or deficient conditions were harvested, and subsequently germinated and grown for 10 days under Pi sufficient conditions (*Figure 1A*), before performing whole genome bisulfite sequencing on the corresponding root genomic DNA (n = 5 for each condition, *Supplementary file 1*). Analysis of the methylation levels in the progeny of stressed and non-stressed parents in the 175 root PSI DMRs that were identified in the first generation of plants failed to identify any significant differences in DNA methylation levels (t-test, FDR < 0.05) (*Figure 4C,D,E*, *Figure 4—figure supplement 2*, Figure 4—source data 3, available at Dryad, *Secco et al., 2015*). Indeed, independent of the treatment performed in the parental generation, DNA methylation levels were reset to a Pi-sufficient like level in the progeny (*Figure 4C,D,E*, *Figure 4—figure supplement 2*). While one DMR (SPX2_DMR2) showed a difference in methylation level between the progeny of plants continuously grown under −P compared to the progeny of those continuously grown in +P, the difference was not statistically significant (t-test, FDR < 0.05). Therefore, it does not appear that these stress induced differential methylation states can be transmitted through meiosis.

Thus, phosphate deprivation induces differential methylation in a variety of plant tissues in TEs that are close to PSI genes, and a limited subset of them, including DMRs associated with key regulators of

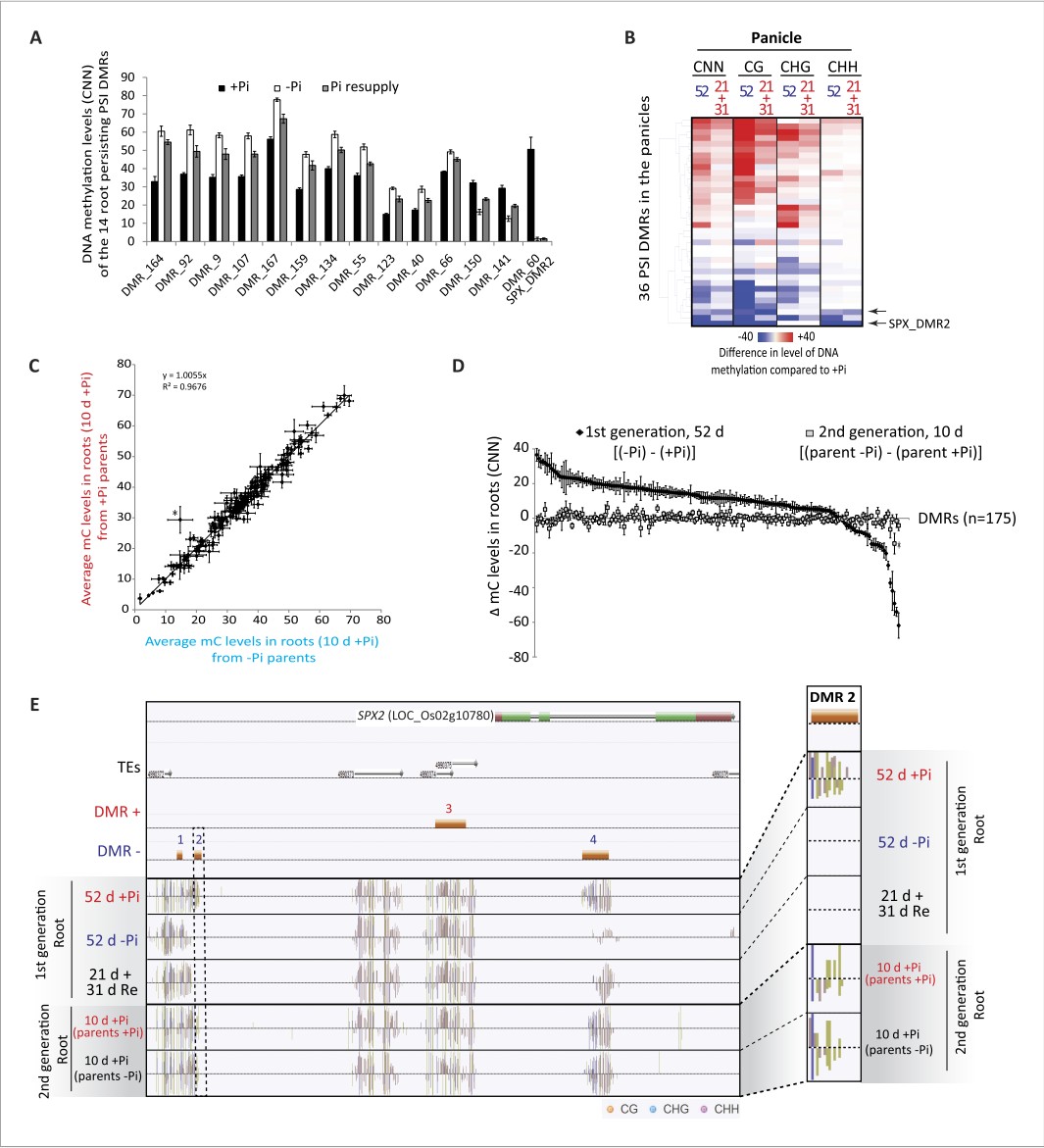

**Figure 4**. Phosphate starvation induced DMRs are mainly transient. (**A**) Methylation levels (CNN) of the 14 regions showing persisting changes in DNA methylation despite 31 days of Pi resupply. Error bars indicates standard error. (**B**) Hierarchical clustering of the difference in methylation level of the 36 DMRs induced by 52 days of Pi deprivation in the panicles. Arrows indicate significant persisting changes in DNA methylation (t-test, FDR < 0.1). (**C**) Scatter plots representing the DNA methylation levels of the 175 PSI DMRs in the progeny of Pi deprived and non-stressed parents. Errors bars indicate standard error (n = 5). A linear trendline as well as its equation is shown. (**D**) Graphical representation of the changes in DNA methylation (CNN) of the 175 PSI DMRs between the first generation after 52 days of Pi treatment [(52d −Pi) − (52d +Pi)] and the offspring of stressed and non-stressed parents grown for 10 days in +Pi. PSI DMRs are sorted based on the change in DNA methylation levels at 52 days, in the first generation. Asterisks represent SPX_DMR2, and error bars indicate standard error. (**E**) Genome browser screenshot of the *SPX2* locus, showing the methylation levels in the first generation at 52 days, as well as in the second generation at 10 days. Source data for *Figure 4* is available at Dryad (*Secco et al., 2015*).

The following figure supplements are available for figure 4:

**Figure supplement 1**. Mitotic transmission of Pi starvation-induced changes in DNA methylation.

**Figure supplement 2**. Hierarchical clustering of the methylation levels (CNN context) of the 175 PSI DMRs in the progeny of stressed and non-stressed parents.

Pi homeostasis such as *SPX2*, can be mitotically transmitted to newly generated cells, despite an extended period of stress recovery. However, no evidence of transgenerational transmission of PSI changes in DNA methylation was observed.

## Stress-induced DMRs are mainly independent of the canonical RdDM pathway

In order to shed light on the DNA methylation pathway mediating Pi starvation-induced changes in DNA methylation, the phosphate starvation experiment was repeated using an RNAi line that knocks down *DCL3a,* a key factor involved in the canonical RdDM pathway. In rice, there are two *DCL3* genes, *DCL3a* and *DCL3b,* the latter also being known as *DCL5* (*Fei et al., 2013*). While *DCL3b* is mainly involved in the generation of stamen-specific 24-nt phased small RNAs, *DCL3a* is involved in producing 24-nt centromere-associated OsCentO siRNAs, MITE-derived siRNAs for abiotic stress responses, and non-canonical long miRNAs (*Wu et al., 2010*; *Yan et al., 2011*; *Song et al., 2012*; *Wei et al., 2014*). Furthermore, it has been shown that reducing the transcription of *DCL3a* via RNAi resulted in more than 80% of all 24-nt clusters being reduced by more than 3 fold, compared to WT (*Wei et al., 2014*). In this study, WT and *DCL3a* RNAi plants were grown as previously described (*Figure 1A*) and subjected to 21 days of Pi sufficient or deficient conditions, before performing RNA-seq and whole genome bisulfite sequencing on the root tissues. RNA-seq analysis (n = 3) confirmed that *DCL3a* was the most abundant *DCL3* family member in the roots, with 11 times higher steady state transcript abundance than *DCL3b* under Pi sufficient conditions (*Figure 5—figure supplement 1A*; Figure 5—source data1, available at Dryad, *Secco et al., 2015*). In the *DCL3a* RNAi line, the transcript abundance of *DCL3a* decreased by 4.6 and 3.3 fold compared to WT in Pi sufficient or deficient conditions, respectively, while the abundance of *DCL3b* transcript was unaffected (*Figure 5—figure supplement 1A,B*, Figure 5—source data1, available at Dryad, *Secco et al., 2015*). In +Pi conditions, the reduced transcription of *DCL3a* was accompanied by widespread changes in DNA methylation, significantly altering DNA methylation levels in 9379 regions in the genome (6694 hypo- and 2685 hypermethylated regions in *DCL3a* RNAi line compared to WT; FDR < 0.05), in addition to 3531 genes displaying a significant change in transcript abundance (Cuffdiff, FDR < 0.05) compared to WT +Pi (*Figure 5A*, *Figure 5—figure supplement 1B*). Similar results were also observed between WT and *DCL3a* RNAi plants under Pi deficient conditions, with 8164 DMRs and 1791 differentially expressed genes identified (*Figure 5A*, *Figure 5—figure supplement 1C*, Figure 5—source data 1, available at Dryad, *Secco et al., 2015*). Analysis of the DMRs induced in *DCL3a* RNAi plants revealed that the majority of changes in DNA methylation occurred in the CG and CHG contexts, under both Pi sufficient and deficient conditions (*Figure 5—figure supplement 1D,E,F,G*). Among the 175 PSI DMRs identified previously (*Figure 2A*), only 8 and 5 regions also showed significant differences in DNA methylation between WT and *DCL3a* plants under Pi sufficient and deficient conditions, respectively, and corresponded to 11 unique PSI DMRs (8 unique associated genes) (*Figure 5A,B,C, D*; Figure 5—source data 2, available at Dryad, *Secco et al., 2015*). These 11 DMRs corresponded to 7 and 4 regions previously identified as hypo- and hypermethylated in response to −Pi in WT, respectively (*Figure 2A*). Of the 4 hypermethylated PSI DMRs altered in the *DCL3a* RNAi plants, only one was associated with significant changes in nearby gene transcript abundance in response to −Pi in WT, while all the hypomethylated PSI DMR associated genes were significantly differentially regulated by −Pi in WT (Figure 5—source data 2, available at Dryad, *Secco et al., 2015*). Furthermore, while reduction in *DCL3a* transcript levels only moderately affected DNA methylation in 6 of the 11 PSI DMRs (mC change < 15%), DNA methylation was almost completely abolished in all contexts in five of these hypomethylated PSI DMRs, which were associated with *SPX1*, *SPX2* and *MGD2* (*Figure 5D,E,F*). Among these, three DMRs were associated with *SPX2*, and showed an average reduction in DNA methylation levels from 47% in WT +Pi to 1% in *DCL3a* RNAi plants +Pi, indicating a requirement for *DCL3a* in maintaining the methylation in these regions (*Figure 5E,F*). Furthermore, among the 8 genes associated with *DCL3a*-dependant PSI DMRs, only *SPX2* showed significant changes in transcript abundance between WT and *DCL3a* RNAi plants, decreasing by 69 fold under +Pi conditions (15.9 FPKM in WT +Pi and 0.2 FPKM in *DCL3a* RNAi plants +Pi) (*Figure 5—figure supplement 1H*). Despite a 2 fold difference in transcript abundance, no significant difference in *SPX2* transcript abundance was observed between WT and *DCL3a* RNAi plants under −Pi conditions, suggesting that the Pi responsiveness of *SPX2* is maintained in the *DCL3a* RNAi plants.

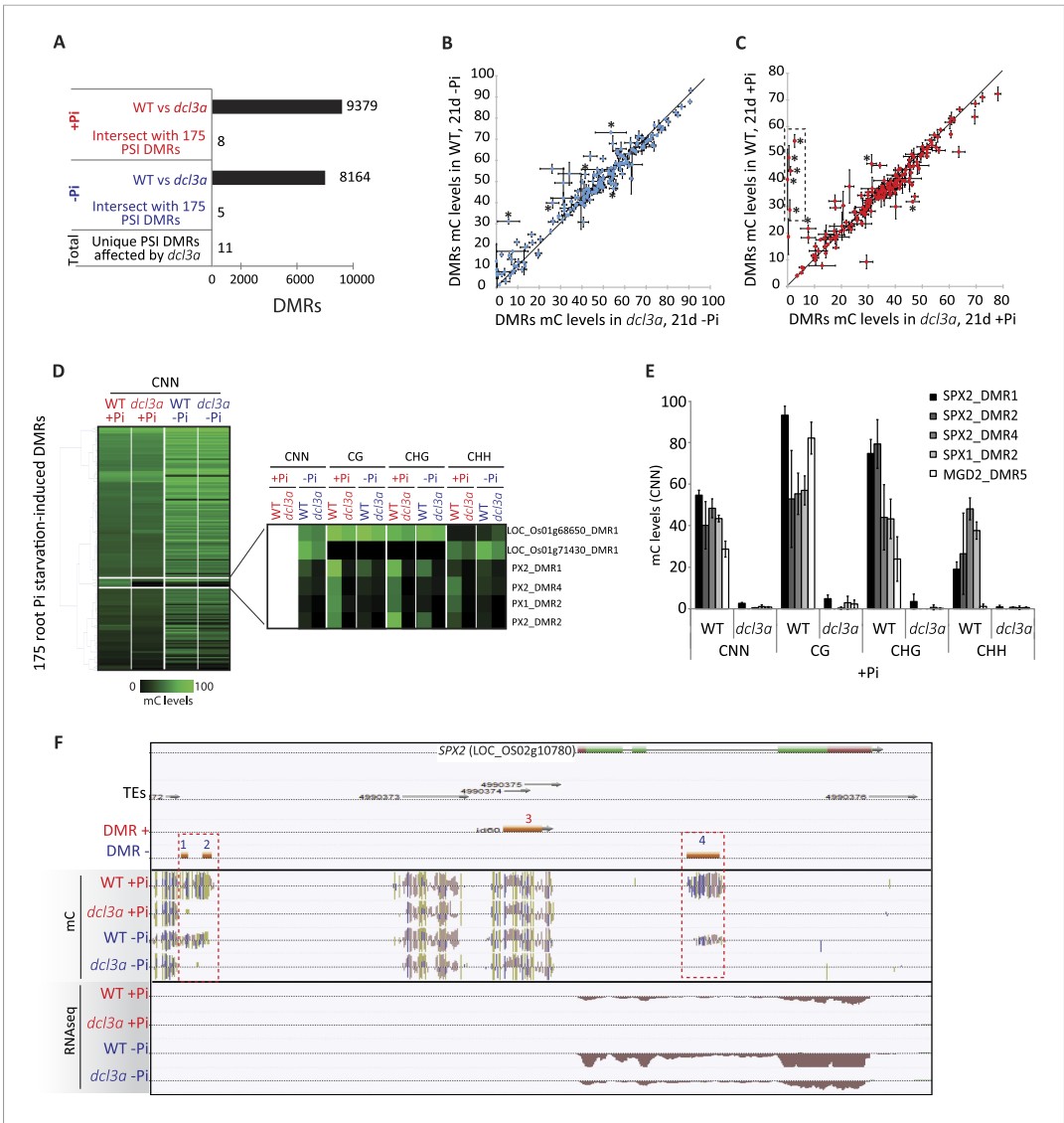

**Figure 5**. *DCL3a* knockdown has limited effects on Pi starvation-induced DMRs. (**A**) Summary of the number of DMRs identified by comparing WT and *DCL3a* RNAi line (referred to as *dcl3a*) root methylomes (21 days) under + and −Pi conditions. For each condition, the *DCL3a* RNAi line-induced DMRs were intersected with the 175 root PSI DMRs, revealing a total of 11 unique PSI DMRs that are significantly altered in *DCL3a* RNAi plants. (**B** and **C**) DNA methylation levels in the 175 PSI DMRs in the *DCL3a* RNAi line vs WT under −Pi and +Pi conditions, respectively. Asterisks represent the significant changes (FDR < 0.05) in DNA methylation levels between WT and the *DCL3a* RNAi line. Black box indicates regions for which the *DCL3a* RNAi line almost completely abolishes DNA methylation, compared to WT. (**D**) Hierarchical clustering of the DNA methylation levels in the 175 PSI DMRs in the *DCL3a* RNAi line vs WT under −Pi (blue) and +Pi (red) conditions. (**E**) DNA methylation levels in 5 regions significantly affected in the *DCL3a* RNAi line. (**F**) Genome browser representation of the *SPX2* locus, showing three *DCL3a*-dependant methylation regions, also responsive to Pi starvation, indicated by red boxes. Source data for *Figure 5* is available at Dryad (*Secco et al., 2015*).

The following figure supplements are available for figure 5:

**Figure supplement 1**. Down-regulation of *DCL3a*.

**Figure supplement 2**. Identification of PSI DMRs in the *DCL3a* RNAi line.

In addition, in order to determine whether *DCL3a* is required for the stability of methylation at a distinct set of genomic loci upon Pi starvation, assessment of changes in DNA methylation between *DCL3a* RNAi plants grown in +Pi and −Pi conditions was conducted. Together, 409 PSI DMRs (FDR < 0.05) were identified in the stressed *DCL3a* RNAi plants (*Figure 5—figure supplement 2*, Figure 5—source data 3, available at Dryad, *Secco et al., 2015*). While the majority of changes in DNA methylation in these 409 DMRs were similar between WT and *DCL3a* RNAi plants (*Figure 5—figure supplement 2*), a subset of regions appeared to show changes in the *DCL3a* RNAi plants but not in WT, suggesting that *DCL3a* may mediate the stability of DNA methylation at some regions of the genome under these stress conditions. Associating each DMR to the nearest gene revealed 88 DMRs that were close to a gene that was significantly differentially expressed in WT upon Pi starvation (Cuffdiff, FDR < 0.05). However, these regions showed similar changes in DNA methylation upon Pi starvation in both WT plants and the *DCL3a* RNAi line. Thus, it appears that while *DCL3a* could potentially mediate the DNA methylation state of some genomic regions in response to Pi starvation, none of these were associated with changes in nearby gene expression.

Overall, it appears that down regulation of *DCL3a* has a very limited effect on the methylation of hypermethylated PSI DMRs, while several regions, particularly those hypomethylated in response to −Pi in WT, fully require the presence of a functional *DCL3a*. Altogether, the majority of PSI DMRs appear to be independent of the canonical RdDM pathway and are thus likely regulated via a different pathway that does not require *DCL3a*.

## Pi starvation in *Arabidopsis* results in a limited number of changes in DNA methylation

To determine whether the widespread PSI DMRs observed in rice could also be seen in other plant species, a similar approach was undertaken in *Arabidopsis*. Due to the shorter life cycle of *Arabidopsis* compared to rice, *Arabidopsis* seedlings were germinated and grown either under Pi sufficient (500 µM) or deficient (13 µM) conditions for 10 days, thus corresponding to a similar stress as the long-term Pi starvation performed in rice relative to the lifespan of the plant. In addition, 7 days after germination, half of the Pi starved plants were transferred to Pi sufficient conditions for 3 days to allow recovery. RNA-seq analysis was performed to assess the transcriptional response to the Pi treatments, identifying 4560 genes displaying differential transcript abundance in the roots after 10 days of Pi starvation (Cuffdiff, FDR < 0.05) (*Figure 6A*, Figure 6—source data 1, available at Dryad, *Secco et al., 2015*). Known PSI marker genes such as *SPX3*, *IPS1*, *miR399* and *PHT1;9* showed >100 fold higher transcript abundance upon Pi starvation. In addition, resupplying Pi for 3 days was sufficient for the transcript abundance of the majority of the PSI genes to return to +Pi like levels. Indeed, of the 4560 PSI genes that displayed differential transcript abundance by 10 days of −Pi, only 10% showed significantly different transcript abundance after 3 days of resupply compared to +Pi. Together, this indicates that the *Arabidopsis* plants responded to both Pi deprivation and Pi resupply.

To examine the extent of DNA methylation changes in response to Pi starvation and recovery in *Arabidopsis*, high coverage whole genome bisulfite sequencing (91–96% cytosines covered, 92–96% of genome covered) was performed on root samples pooled from 10 plants, in triplicate (*Supplementary file 1*). While no PSI DMRs could be detected using a FDR < 0.05, two PSI DMRs were identified using lower stringency parameters (FDR < 0.2), requiring only two of the three replicates to show significant methylation differences for each condition. Assignment of these DMRs to the nearest gene revealed that they were associated with two known PSI genes, *MGD3*, encoding for the major enzyme for galactolipid metabolism during phosphate starvation (*Kobayashi et al., 2009*), and *Atcopeg1* (Copia evolved gene 1), the only expressed gene derived from the AtCopia95 retrotransposon in the *Arabidopsis* genome, which has been shown to be involved in many developmental and adaptive processes, including Pi starvation (*Duan et al., 2008*) (*Figure 6B,D*). Both DMRs overlapped with TEs and were hypermethylated in response to −Pi, preferentially in the CHH context (*Figure 6B,C,D,E*). For both DMRs, resupplying Pi for 3 days resulted in a significant reduction in the CHG and CHH methylation levels compared to −Pi (*Figure 6C,E*). For example, while under Pi sufficient conditions only 10% of the cytosines in the CHH context were methylated in the DMR associated with *Atcopeg1*, Pi starvation increased the number of methylated cytosines in the CHH context to 40%, before returning to 24% upon 3 days of Pi resupply, suggesting that the changes in DNA methylation

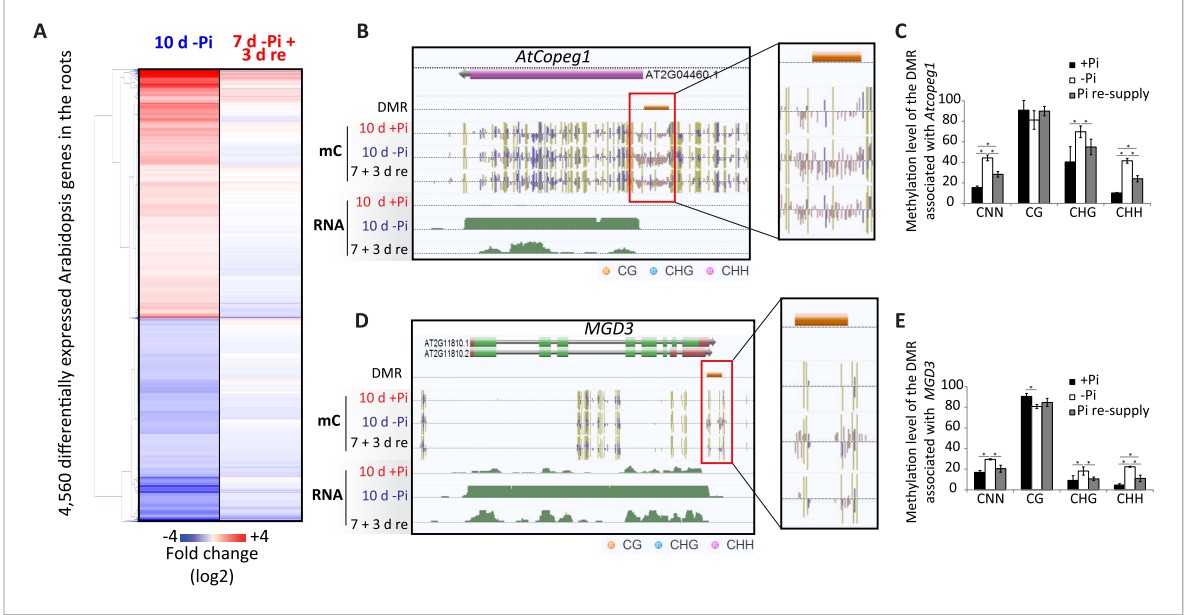

**Figure 6**. Pi starvation in *Arabidopsis* results in a limited number of changes in DNA methylation. (**A**) Hierarchical clustering of significantly differentially regulated genes upon 10 days of Pi starvation and resupply. *Arabidopsis* seeds were germinated and grown in either Pi sufficient (500 μM Pi) or deficient (13 μM Pi) for 10 days. After 7 days of treatment, half of the Pi starved plants were re-supplied with 500 μM Pi for 3 days. (**B** and **C**) Genome browser representation of the two Arabidopsis PSI DMRs identified in this study (FDR < 0.2). *AtCopeg1* (Copia evolved gene 1) is a Copia like transposon previously shown to be induced by Pi starvation (*Duan et al., 2008*), while *MGD3* (monogalactosyl diacylglycerol synthase 3) encodes the major enzyme for galactolipid metabolism during Pi starvation. (**C** and **E**) DNA methylation levels in the CNN, CG, CHG and CHH contexts in the three Pi treatments (+Pi, −Pi and Pi resupply) in the DMRs associated with *Atcopeg1* and *MGD3*, respectively. Asterisks represents significant differences in the methylation level (t-test < 0.05). Source data for *Figure 6* is available at Dryad (*Secco et al., 2015*).

observed are transient. Overall, despite Pi starvation having a limited effect upon DNA methylation in *Arabidopsis* compared to rice, with only 2 PSI DMRs identified, the location of these PSI DMRs appears to be conserved, with hypermethylated regions overlapping with specific TEs that are close to highly induced PSI genes.

## Discussion

Plants have developed a wide array of mechanisms to tolerate changing environmental conditions. However, despite our understanding of Pi homeostasis greatly increasing during the past decade, very little is known about the potential role of changes in the epigenome in response to Pi starvation, and to stresses in general. Comprehensive methylome profiling coupled with an extensive time course experiment enabled investigation of the presence and potential role of dynamic epigenomic changes upon stress, revealing the temporal relationship between changes in DNA methylation and transcription, and the stability of altered DNA methylation states through mitosis and meiosis. Indeed, this study provides insights into the dynamic changes in DNA methylation in response to Pi starvation, with the identification of widespread changes in DNA methylation near PSI genes, in a species-dependent manner, likely correlated with the TE content of the genome. It is well known that plant genomes exhibit a very large diversity of size and repeat content as well as methylation levels, which were recently shown to be positively correlated, thus highlighting the central role of DNA methylation in repressing the potentially deleterious activity of TEs through hypermethylation (*Mirouze and Vitte, 2014*). However, the majority of DNA methylation analyses performed in plants to date have focused on *Arabidopsis*, despite being relatively depleted of TEs (15–20% of the genome) and being poorly methylated compared to other plant genomes (*Mirouze and Vitte, 2014*). Potentially, the differences in DNA methylation changes observed in rice and *Arabidopsis* in response to Pi starvation could primarily be due to the differential frequency of TEs in the genome of each species.

To date, only a limited number of studies have comprehensively investigated the involvement of DNA methylation in response to adverse environmental conditions (*Dowen et al., 2012*, *Liang et al., 2014*). Several studies have reported that changes in the environment can affect the methylation status of some regions of the genome, using low resolution and non-quantitative techniques such as methylation-sensitive amplification polymorphism (*Choi and Sano, 2007*; *Hauser et al., 2011*; *Wang et al., 2011a*, *2011b*, *Karan et al., 2012*; *Shaik and Ramakrishna, 2012*; *Sahu et al., 2013*; *Yu et al., 2013*). However, these studies have lacked the resolution to provide the specific context and genomic location of the changes in DNA methylation, thus offering limited insights into the potential role of stress-induced changes in DNA methylation. While this study primarily focused in the main organ involved in nutrient sensing and uptake, namely the root, both roots and shoots appeared to exhibit similar changes in DNA methylation in response to Pi starvation. In addition, all but one of these DMRs overlapped with TEs. While, several studies have suggested that there is a negative relationship between methylation of TEs and nearby gene transcription (*Hollister and Gaut, 2009*; *Ahmed et al., 2011*; *Eichten et al., 2012*), we observed the opposite trend, with hypermethylated DMRs being located in close proximity to PSI genes. Consequently, we sought to shed light upon the temporal hierarchy, and thus potential causal relationships, between the observed environmentally-induced differential DNA methylation and the changes in gene transcript abundance. Using our time-course experiment, we demonstrated that upon Pi stress, changes in transcription occurred prior to changes in DNA methylation levels, thus suggesting that induction of gene transcription may be causal for the differential DNA methylation. In addition, since almost all the regions gaining DNA methylation in response to Pi deprivation overlap with TEs, such a process could potentially constitute a mechanism to repress the activity of specific TEs that are near highly induced PSI genes when plants are confronted with −Pi conditions. Indeed, vonHoldt and colleagues recently showed that the methylation pattern of TEs is a complex function of TE size, age and distance to a gene, with young TEs being preferentially poorly methylated (*vonHoldt et al., 2012*). During Pi stress, PSI genes are highly expressed, presumably requiring increased chromatin accessibility to facilitate access of Pol II and other transcription machinery, which may facilitate the transcription of poorly methylated nearby TEs, with potentially deleterious effects upon the genome such as insertional inactivation of genes and ectopic recombination. This hypothesis is strengthened by the fact that the majority of the hypermethylated PSI DMRs are independent of *DCL3a*, a key component of the canonical RdDM pathway. Thus, we hypothesize that the RDR6-RdDM pathway, another entry point into RdDM that recognizes Pol II-derived TE transcripts, may act to methylate these TEs. Overall, the observed increase in DNA methylation in specific TEs located close to PSI genes provides insights into a cellular activity that may act to maintain localized suppression of transposable elements in genomic regions that must be transcriptionally activated to respond to environmental perturbation (*Figure 7*).

In contrast, hypomethylated rice root PSI DMRs overlapped less frequently with TEs, and were often affected by knockdown of *DCL3a*, suggesting a different function of this subset of PSI DMRs, and likely involving the canonical RdDM pathway. Among the 11 *DCL3a*-dependent PSI DMRs, those associated with *SPX1* and *SPX2* were the most affected by reduced transcription of *DCL3a*. It has recently been shown that both SPX1 and SPX2 can inhibit phosphate starvation responses through Pi-dependant direct interaction with PHR2, a key transcription factor controlling the majority of Pi-responsive genes. Under Pi sufficient conditions, SPX1 and SXP2 are tightly bound to PHR2, preventing PHR2 from interacting with its cognate binding sites, and suppressing induction of Pi responsive genes (*Puga et al., 2014*, *Wang et al., 2014*). Surprisingly, *SPX2* transcript abundance was >65 fold lower in the *DCL3a* RNAi line compared to WT, potentially enabling PHR2 to interact with its target binding sites and induce transcription of PSI genes. Such a mechanism would thus explain the high number of differentially expressed genes observed between WT and the *DCL3a* RNAi line under +Pi conditions (3531), compared to −Pi (1791), where *SPX2* is expressed at similar levels between WT and the *DCL3a* RNAi line. However, it remains unclear how the knockdown of *DCL3a* can repress *SPX2* to such an extent in +Pi conditions. It is unlikely that the repression of *SPX2* is a direct consequence of *DCL3a*-dependent changes in DNA methylation at this locus, since similar DNA methylation patterns are observed in *DCL3a* RNAi plants under +Pi and −Pi, while *SPX2* is highly induced in −Pi and not expressed in +Pi (*Figure 5F*). In addition, several studies have shown that changes in DNA methylation within or near DNA binding elements could interfere with the binding of cognate transcription factors (*Deng et al., 2001*; *Bird, 2002*; *Zhong et al., 2013*). Zhong and colleagues (2013) recently showed in tomato that binding sites for RIN (Ripening Inhibitor), one the main transcription

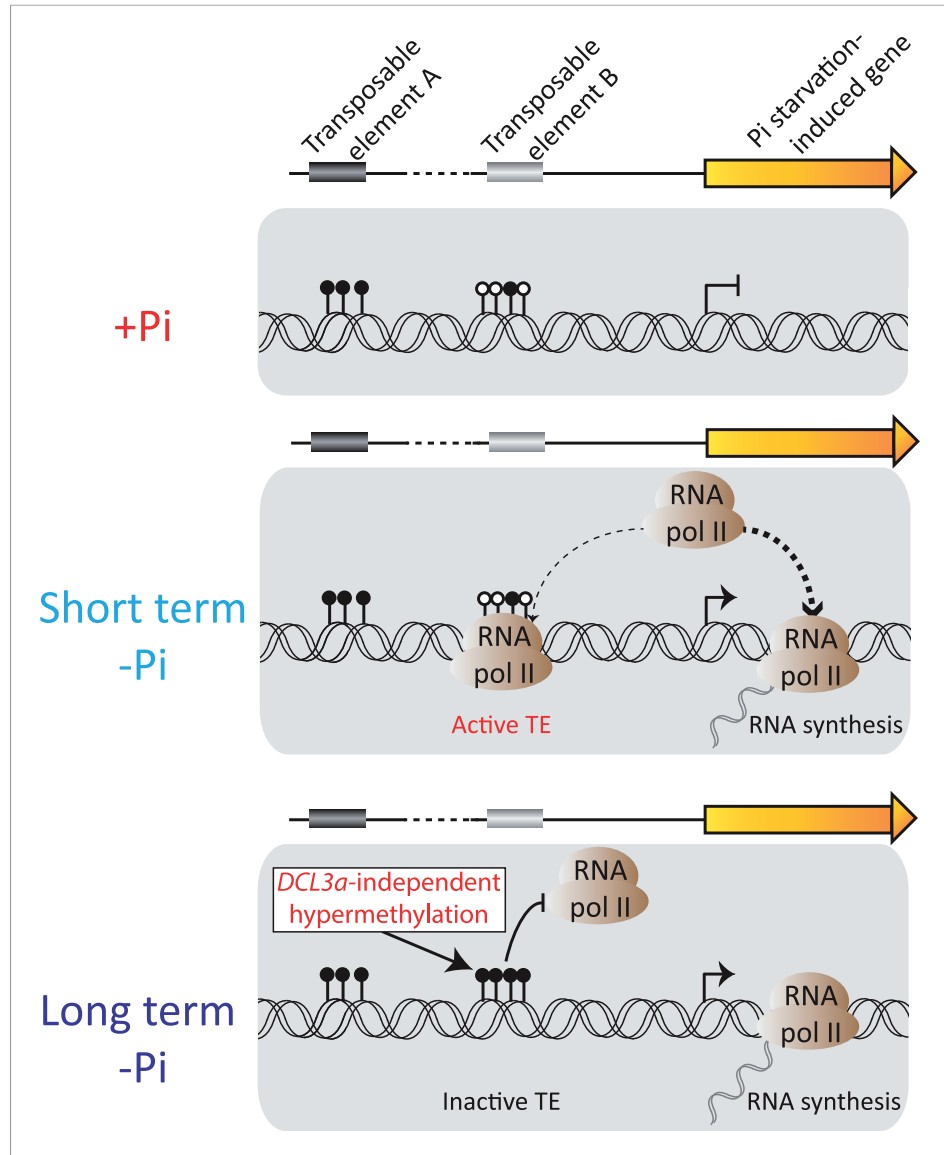

**Figure 7**. Model of the role of DNA methylation in response to Pi starvation. Schematic of a PSI gene associated with two TEs. TE A represents TEs that are located distant from genes and are highly methylated, while TE B represents TEs that are close to genes and lowly methylated. Upon short term Pi deprivation, RNA polymerase II is recruited to Pi starvation-induced (PSI) genes, resulting in increased PSI gene expression. These high levels of RNA polymerase II could induce transcription of nearby and poorly methylated TEs, such as TE B, which could have deleterious effects upon the plant. As a consequence, upon prolonged Pi deprivation, TEs that are close to highly expressed stress-induced genes are hypermethylated, via a *DCL3a*-independent mechanism, thus preventing their transcription via RNA polymerase II. Black and white circles represent methylated and unmethylated cytosines, respectively. Thickness of the dashed lines represents the proportion of RNA polymerase II recruited to PSI genes or TEs.

factors involved in fruit ripening, were frequently demethylated during ripening, thus enabling the induction of ripening genes (*Zhong et al., 2013*). Such a mechanism could potentially occur in rice, fine-tuning the expression of specific genes involved in Pi homeostasis by regulating the binding capacity of specific transcription factors to PSI genes. However, the involvement of such a mechanism in the Pi starvation response remains to be determined.

The Pi resupply experiment allowed us to determine whether PSI DMRs could be maintained despite an extensive period of Pi recovery and tissue growth. While the majority of PSI DMRs returned towards +Pi DNA methylation levels, 14 root PSI DMRs showed significant sustained

changes in DNA methylation despite 31 days of Pi resupply, suggesting that these epialleles are likely to be transmitted to newly generated cells, and can thus be mitotically inherited. In addition, unlike animals where DNA methylation is reset in primordial germ cells and during embryogenesis, DNA methylation states in plants can be stably transmitted from parents to offspring (*Kinoshita et al., 2007*; *Becker et al., 2011*; *Schmitz et al., 2011*; *Weigel and Colot, 2012*), and thus could potentially establish a transgenerational 'memory' of the stress. However, to date few reports have studied the potential heritability of stress-induced DNA methylation changes, and no evidence of transgenerational transmission of stress-induced differential DNA methylation exists (*Hauser et al., 2011*, *Pecinka and Scheid, 2012*; *Sahu et al., 2013*). Boyko and colleagues reported that the progeny of stress-treated plants had increased homologous recombination frequency and global DNA methylation levels, as well as higher tolerance to stress (*Boyko et al., 2010*). However, no evidence of transgenerational transmission of differences in DNA methylation was observed, and the increase in global genome methylation levels did not persist in successive generations during the absence of stress (*Boyko et al., 2010*). Additionally, several recent studies suggest that specific mechanisms exist to prevent transgenerational inheritance of stress-induced epigenetic states (*Baubec et al., 2014*; *Crevillen et al., 2014*; *Iwasaki and Paszkowski, 2014*). Furthermore, Hagmann and colleagues, using a near-clonal North American *A. thaliana* population that has diverged under natural conditions for at least a century, recently showed that environment-induced changes are only minor contributors to durable genome-wide heritable epigenetic variation, with more than 97% of the total methylated genome space not being altered by the environment across dozens of generations (*Hagmann et al., 2015*). In this study, we found no evidence of transgenerational inheritance of PSI DMRs, suggesting that PSI DMRs, and potentially most stress-induced differential methylation in general, are not transmitted to their progeny and are thus not likely to contribute to transgenerational stress memory.

Altogether, this study reveals a species-specific process in which an abiotic stress induces dynamic and widespread changes in DNA methylation in rice. In addition, we establish the temporal relationship between differential transcription and DNA methylation, the limited stability of such induced DNA methylation events through mitosis, and the absence of their transmission through meiosis. These findings have important implications for interpreting the capacity of environmentally induced epialleles to influence genic transcription, and their stability through plant growth and reproduction.

## Materials and methods

### Plant materials and growth conditions

Rice (*Oryza sativa* L. *cv. Nipponbare*) was used for all physiological experiments. Hydroponic experiments were performed under controlled conditions (day/night temperature of 30/22°C and a 12 hr photoperiod, 200 µmol photons $m^{-2}$ $s^{-1}$), allowing 0.5 l of hydroponic solution per plant. The hydroponic solution consisted of a modified solution as described in (*Secco et al., 2013a*), containing 1.425 mM $NH_4NO_3$, 0.513 mM $K_2SO_4$, 0.998 mM $CaCl_2$, 1.643 mM $MgSO_4$, 0.075 µM $(NH_4)_6Mo_7O_{24}$, 0.25 mM $NaSiO_3$, 0.009 mM $MnCl_2$, 0.019 µM $H_3BO_3$, 0.155 µM $CuSO_4$, 0.152 µM $ZnSO_4$ and 0.125 mM EDTA-Fe, with or without 0.323 mM $NaH_2PO_4$, resulting in the +Pi and −Pi conditions. The pH of the solution was adjusted to 5.5 and the solution was renewed every 3 day. Rice seeds were first pre-germinated in tap water for 2 days before being transferred into the hydroponic solution, containing 0.323 mM Pi (+Pi) for 2 weeks. Half of the seedlings were then transferred to a solution lacking Pi (0 mM Pi) for 21 days, before being re-supplemented with 0.323 mM Pi for up to 31 days, while the other half of the seedlings continuously remained in +Pi conditions (control). During the resupply experiment, half of the rice seedlings were left in Pi deficient media, to serve as control. After 24 days of Pi starvation, plants grown under Pi deficient conditions were supplemented with 0.03 mM Pi (1/10th of Pi sufficient Pi concentration) until the end of the experiment to prevent them from dying. Roots and shoots were harvested separately at each time points. Furthermore, all 'Materials and methods' such as media replacement and sample collection were performed at similar time of the day (2 hr after light) in order to minimize possible circadian effect.

For *Arabidopsis*, seeds were germinated and grown vertically on Murashige and Skoog medium diluted 10-fold in Petri dishes supplemented with a Pi source of either 500 µM (+P) or 13 µM (-P) $NaH_2PO4$ in a culture chamber under a 16-hr- light/8-hr-dark regime (25°C/22°C), as previously described (*Misson et al., 2004*).

## Pi content determination

Determination of Pi in tissues was measured by releasing the cellular content of cells into water by repeated freeze–thaw cycle, or by incubation for 1 hr at 85°C, and quantifying Pi by the molybdate assay according to the procedure of Ames (*Ames, 1966*).

## Total RNA isolation and RNA-seq library preparation

The total RNA from the roots and shoots tissues was extracted using TRIzol reagent (Invitrogen, Carlsbad, CA), according to the manufacturer's instructions. For *Arabidopsis*, RNA was extracted using the Spectrum Plant Total RNA kit from Sigma (St Louis, MO). The integrity and quality of the total RNA was determined using NanoDrop 1000 Spectrophotometer and formaldehyde-agarose gel electrophoresis. RNA was only used when the Abs260 nm/Abs280 nm ratio was >1.8.

For RNA-seq library synthesis (3 biological replicates per condition, 1 plant per replicate, except for *Arabidopsis* where only two replicates were used), total RNA was first depleted of rRNA using the Ribo-Zero rRNA removal kit (Plant Leaf, and Plant Seed/Root kits, Epicentre, Madison, WI). To do so, 1 μg of total RNA from root samples was used as input for rRNA removal, while 2 μg of total RNA was used for shoot samples. Sequencing libraries were generated using the TruSeq RNA Sample Prep Kit (Illumina, San Diego, CA).

For the rice 24 day and 52 day time point samples and the *Arabidopsis* samples, the Illumina TruSeq Stranded Total RNA with Ribo-Zero Plant kit was used to generate the libraries (Illumina).

## Genomic DNA extraction and MethylC-Seq library generation

Genomic DNA (gDNA) was extracted plant tissues using the DNeasy Plant minikit (Qiagen, the Netherlands). 600 ng of purified gDNA, spiked in with 0.5% (wt/wt) unmethylated Lambda DNA (Promega, Madison, WI) was used to prepare MethylC-seq libraries using three biological replicates per condition (1 plant per replicate for rice), as previously described (*Lister et al., 2013*). In *Arabidopsis*, due to the reduced size of the roots and thus reduced amount of gDNA recovered, roots of 10 plants were pooled per replicate, and three independent replicates were used per time point. Briefly, gDNA was sonicated (Covaris) to a mean fragment size distribution of ~200 bp. Sheared ends were repaired (End-It DNA End Repair kit, Epicentre, Madison, WI), followed by 3′ A-tailing with 0.2 mM dATP and 15 U of Klenow fragment of DNA polymerase and ligation of single-read methylated adapters (Illumina) using 15 U of T4 DNA ligase (New England Biolabs, Ipswich, MA). Sodium bisulfite conversion was then performed using the MethylCode Bisulfite conversion kit (Life Technologies, Carlsbad, CA) according to the manufacturer's instructions. Libraries were then amplified by PCR using 20 μl of bisulfite converted adapter-ligated DNA molecules were mixed with 25 μl of KAPA HiFi Hotstart Uracil+ ReadyMix (Kapa Biosystems, Wilmington, MA) and 5 μl of Truseq Primer Cocktail (Illumina) with cycle conditions of an initial denaturing at 95°C for 2 min, followed by 6 cycles of 98°C for 20 s, 60°C for 15 s, and 72°C for 1 min, followed by 10 min at 72°C. Libraries were then purified with AMPure XP beads (Beckman Coulter, Brea, CA) before being sequenced for 101 cycles using the Illumina HiSeq 1500, as per manufacturer's instructions.

## Mapping of RNA-Seq reads, transcript assembly, and abundance estimation using tuxedo suite

All RNA-seq analyses were performed as previously reported (*Secco et al., 2013a*). Briefly TopHat2 and Cufflinks2 (*Trapnell et al., 2012*, *2013*) packages were used to map sequence reads to the rice IRGSP-1.0 and *Arabidopsis* TAIR10 reference genomes and quantitate differential gene transcription (FDR < 0.05). TEs were classified using the classification of Plant Repeat database (http://plantrepeats. plantbiology.msu.edu/about.html#codes).

## MethylC-seq data analysis

Read mapping, processing, and analysis were performed as described previously (*Lister et al., 2013*), aligning reads to the rice IRGSP-1.0 and *Arabidopsis* TAIR10 reference genomes. To estimate the bisulfite non-conversion frequency, the frequency of all cytosine basecalls at reference cytosine positions in the unmethylated control lambda genome was normalized by the

total number of basecalls at cytosine reference positions in the lambda genome. DMRs were identified as previously described (*Lister et al., 2013*). Briefly, for each cytosine in the CNN context, a root mean square test was performed as previously reported (*Perkins et al., 2011*). To do so, a contingency table where rows indicated the position of each cytosine in the CNN context on the genome and the columns indicated the number of reads that supported a methylated cytosine or an unmethylated cytosine was generated. Next, p-values were simulated using 10,000 permutations and for each permutation, a new contingency table was generated by randomly assigning reads to cells with a probability equal to the product of the row marginal and column marginal divided by the total number of reads squared. To increase the efficiency of this process, the permutations were stopped when a p-value returned 100 permutations with a statistic greater than or equal to the original test statistic. The p-value cutoff that would control the FDR at our desired rate was determined as previously reported (*Bancroft et al., 2013*). The largest p-value cutoff that still satisfied our FDR requirement was then chosen and significant sites showing changes in DNA methylation were combined into DMRs if they were within 200 bases of one another and had methylation changes in the same direction. Furthermore, blocks that contained fewer than 8 and 5 differentially methylated sites were discarded in the rice and *Arabidopsis* analyses, respectively. The sample comparison details for the identification of the DMR sets described in this study are described below.

- *Rice root +Pi* vs *−Pi DMRs* were identified (FDR < 0.01) between (21, 22, 24 days) +Pi and (21, 22, 24 days) −Pi samples, requiring ≥7 of 9 root −Pi samples to be differentially methylated compared to root + Pi samples, as well as enabling 1 of 9 root −Pi samples to be similar to + Pi and *vice versa*.
- *Rice shoot +Pi* vs *−Pi DMRs* were identified (FDR < 0.05) between 21 days +Pi and 21 days −Pi samples, requiring 3 of 3 −Pi shoot samples to be differentially methylated compared to +Pi shoot samples.
- *Arabidopsis root +Pi* vs *−Pi DMRs* were identified (FDR < 0.2) between 10 days +Pi and 10 days −Pi samples, requiring 3 of 3 −Pi root samples to be differentially methylated compared to +Pi root samples.

Persisting differences in DNA methylation levels were defined as being significantly different between 52 days + Pi and 52 days −Pi conditions (t-test, FDR < 0.05) and significantly different between 52 days Pi resupply and 52 days +Pi (t-test, FDR < 0.05), but not significant different between 52 days −Pi and 52 days Pi resupply (t-test, FDR >0.05). Persisting DMRs were identified in CNN context. FDR was calculated using the Benjamini Hochberg method.

Full browsing of the entire rice and *Arabidopsis* datasets can be found at http://www.plantenergy.uwa.edu.au/public/annoj/rice_Pi_starved_methylomes_Secco_et_al.htm and http://www.plantenergy.uwa.edu.au/public/annoj/Arabidopsis_Pi_starved_methylomes_Secco_et_al.htm, respectively. Transposable element track was obtained from Oryza Repeat Database (http://rice.plantbiology.msu.edu/annotation_oryza.shtml).

## Estimation of methylation level (mC/C) with correction for non-conversion

To calculate the methylation levels (mCNN/CNN, mCG/CG, mCHG/CHG or mCHH/CHH), which is an estimate of the fraction of cytosines in the sequenced population which are methylated, we computed the fraction of all MethylC-Seq basecalls at cytosine reference positions that were cytosine (protected from bisulfite conversion), and then subtracted these estimates for the failure of the chemical conversion of unmethylated cytosine (non-conversion rate), based on the non-conversion rate to the unmethylated Lambda DNA spiked-in control.

## Accession numbers

Illumina reads of all samples have been submitted to the Sequence Read Archive at the National Center for Biotechnology Information (http://www.ncbi.nlm.nih.gov/sra) under accession number SRP061678 (Stranded RNA-seq for rice after 3, 24 and 52 days of Pi stress), SRP032765 (Rice methylomes under Pi stress), SRP061677 (Stranded RNA-seq and methylomes for the rice DCL3a RNAi experiment), and SRP040029 (Methylomes and stranded RNA-seq for Arabidopsis under Pi stress).

Source data are deposited in the DRYAD repository: 10.5061/dryad.40gd6. (*Secco et al., 2015*).

## Acknowledgements

The authors would like to thank Robert Schmitz and Justin Borevitz for critical reading of the article and useful discussions, as well as Xiaofeng Cao for sharing the *DCL3a* RNAi line.

## Additional information

### Funding

| Funder | Grant reference | Author |
| --- | --- | --- |
| Gordon and Betty Moore Foundation | GBMF3034 | Joseph R Ecker |
| Australian Research Council (ARC) | FT120100862 | Ryan Lister |
| Australian Research Council (ARC) | FS100100022 | David Secco |
| Australian Research Council (ARC) | DE150100460 | David Secco |
| Australian Research Council (ARC) | CE140100008 | James Whelan, Ryan Lister |

The funders had no role in study design, data collection and interpretation, or the decision to submit the work for publication.

### Author contributions

DS, JW, Conception and design, Acquisition of data, Analysis and interpretation of data, Drafting or revising the article; CW, SC, Acquisition of data; HS, Acquisition of data, Drafting or revising the article; MDS, JRE, Analysis and interpretation of data; LN, Acquisition of data, Analysis and interpretation of data; RL, Conception and design, Acquisition of data, Analysis and interpretation of data, Drafting or revising the article

## Additional files

### Supplementary file

• Supplementary file 1. Summary statistics of MethylC-seq reads.

### Major datasets

The following datasets were generated:

| Author(s) | Year | Dataset title | Dataset ID and/or URL | Database, license, and accessibility information |
| --- | --- | --- | --- | --- |
| Secco D, Wang C, Shou H, Schultz M, Chiarenza S, Nussaume L, Ecker JR, Whelan J, Lister R | 2015 | Stranded RNA-seq for rice after 3, 24 and 52 days of Pi stress | http://www.ncbi.nlm.nih.gov/sra/?term=SRP061678 | Publicly available at the NCBI Short Read Archive (Accession no: SRP061678). |
| Secco D, Wang C, Shou H, Schultz M, Chiarenza S, Nussaume L, Ecker JR, Whelan J, Lister R | 2015 | Rice methylomes under Pi stress | http://www.ncbi.nlm.nih.gov/sra/?term=SRP032765 | Publicly available at the NCBI Short Read Archive (Accession no: SRP032765). |
| Secco D, Wang C, Shou H, Schultz M, Chiarenza S, Nussaume L, Ecker JR, Whelan J, Lister R | 2015 | Methylomes and stranded RNA-seq for Arabidopsis under Pi stress | http://www.ncbi.nlm.nih.gov/sra/?term=SRP040029 | Publicly available at the NCBI Short Read Archive (Accession no: SRP040029). |
| Secco D, Wang C, Shou H, Schultz M, Chiarenza S, Nussaume L, Ecker JR, Whelan J, Lister R | 2015 | Stranded RNA-seq and methylomes for the rice DCL3a RNAi experiment | http://www.ncbi.nlm.nih.gov/sra/?term=SRP061677 | Publicly available at the NCBI Short Read Archive (Accession no: SRP061677). |

| Author(s) | Year | Dataset title | Dataset ID and/or URL | Database, license, and accessibility information |
|---|---|---|---|---|
| Secco D, Wang C, Shou H, Schultz M, Chiarenza S, Nussaume L, Ecker JR, Whelan J, Lister R | 2015 | Data from: Phosphate starvation induces widespread modulation of DNA methylation at stress responsive genes | http://datadryad.org/review?doi=doi:10.5061/dryad.40gd6 | Available at Dryad Digital Repository under a CC0 Public Domain Dedication. Figure 1—source data 1: FPKM and fold change of all genes in rice roots and shoots upon Pi treatments. Figure 2—source data 1: List of 175 PSI DMRs identified in the root, with DNA methylation levels and transcription levels of DMR-associated genes. Figure 2—source data 2: Transcription levels of TEs overlapping PSI DMRS after 7 days and 21 days of Pi deprivation. Figure 2—source data 3: List of 341 PSI DMRs identified in the shoots, with DNA methylation levels and transcript abundance levels of DMR-associated genes. Figure 3—source data 1: Levels of DNA methylation and FDR values for the 100 PSI DMRs associated with changes in nearby gene transcript abundance. Figure 4—source data 1: List of 14 root persisting PSI DMRs, despite 31 days of Pi resupply. Figure 4—source data 2: List of 36 panicle PSI DMRs. Figure 4—source data 3: Average methylation levels observed in the roots of the progeny of stressed- and non-stressed plants in all contexts and methylation levels for each replicate in the CNN context in the 175 root PSI DMRs. Figure 5—source data 1: FPKM and fold change of all genes between WT +Pi and DCL3a RNAi line +Pi as well as WT −Pi and DCL3a RNAi line −Pi (FDR < 0.05). Figure 5—source data 2: 11 DCL3a-dependant PSI DMRs, with methylation levels and associated gene transcript abundance. Figure 5—source data 3: 409 PSI DMRs in the DCL3a RNAi line, and 88 PSI DMRs in the DCL3a RNAi line, associated with |

| Author(s) | Year | Dataset title | Dataset ID and/or URL | Database, license, and accessibility information |
|---|---|---|---|---|
| | | | | significant changes in nearby gene expression. Figure 6—source data 1: FPKM and fold change of all genes in Arabidopsis roots upon Pi starvation. |

The following previously published dataset was used:

| Author(s) | Year | Dataset title | Dataset ID and/or URL | Database, license, and accessibility information |
|---|---|---|---|---|
| Secco D, Jabnoune M, Walker H, Shou H, Wu P, Poirier Y, Whelan J | 2013 | RNA-seq data for rice after 7, 21, 22 days of Pi stress | http://www.ncbi.nlm.nih.gov/sra/?term=SRA097415 | Publicly available at the NCBI Short Read Archive (Accession no: SRA097415). |

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
