## [Decision Letter]

Thank you for submitting your work entitled “Phosphate starvation induces widespread modulation of DNA methylation at stress responsive genes” for further consideration at *eLife*. Your article has been favorably evaluated by Detlef Weigel (Senior editor and Reviewing editor), and three reviewers. The manuscript has been improved since the previous version but there are some remaining issues that need to be addressed before final acceptance.

The reviewers very much appreciated your diligent follow up, and agreed that the observation of TE methylation following transcription of adjacent genes is an important one. Not only does it go against the commonly accepted wisdom of transcription being an outcome in changes of methylation, but your demonstration that this mostly does not involve the canonical RdDM pathway may also reveal a formerly unappreciated route to (somatic) methylation changes.

We would ask you for some minor tweaks. First, it would be good if the title better reflected the new data, and that you refer to the temporal order of transcription and methylation changes. Second, the reviewers were wondering whether there are phosphate-starvation-induced DMRs between DCL3a control and DCL3a stressed plants. The question that could be addressed with this is whether DCL3a is required for the stability of other regions in the genome that might not be the same as the stress-induced ones in the wild type. Finally, there are some minor changes, as listed below.

Abstract

Abstract. I find the word “proximal” a bit confusing. I would propose “nearby” or “close”.

In the first paragraph of the subsection “Pi starvation induces widespread changes in DNA methylation, enriched at key regulators of Pi homeostasis”. I could not find how many ready had to cover a position in order to be considered as covered. This should be specified.

The statement “the root PSI DMRs had an average size of 205 bp with a mean of 16 differentially methylated cytosines per DMR, and were preferentially localized within the first kilobase upstream (20%) and downstream (20%) of the proximal gene (Figure 2, Figure 2—figure supplement 2)” and Figure 2 may be potentially wrong. There are many more short intergenic regions than the long ones, which typically leads to the pattern described by the authors. In order to perform this analysis in an unbiased way, the authors should calculate this as percentage within individual intergenic regions.

Discussion: “the differences in DNA methylation changes observed in rice and *Arabidopsis* in response to Pi starvation could primarily be due to the differential frequency of TEs in the genome of each species, suggesting that *Arabidopsis* may not be the best model in which to study the role of DNA methylation in controlling TE activity in plants”. I think the large number of very successful genetic screens with results published in the top journals disproves this statement.

Discussion: “Thus, the RDR6-RdDM pathway, another entry point into RdDM that recognizes Pol II-derived TE transcripts, may act to methylate these TEs. The authors should include “we hypothesize”.

[Editors’ note: a previous version of this study was rejected after peer review, but the authors submitted for reconsideration. The previous decision letter after peer review is shown below.]

Thank you for choosing to send your work entitled “Phosphate starvation in rice induces mitotically heritable modulation of DNA methylation at starvation responsive genes” for consideration at *eLife*. Your full submission has been evaluated by Detlef Weigel (Senior editor and Reviewing editor) and three peer reviewers, and the decision was reached after discussions between all involved. Based on our discussions and the individual reviews below, we regret to inform you that your work will not be considered further for publication in *eLife* at this time.

Detlef Weigel and the reviewers had a long discussion about the merits of the work. On the plus side, there was broad agreement that this is likely the most rigorous study of stress-induced DNA methylation changes in plants. On the negative side, it is difficult to come to any strong conclusions. First, the question of whether methylation changes are mostly a consequence of altered transcription or vice versa is not really solved. Much of the evidence points to transcriptional changes being more important, but there is some tantalizing evidence for the opposite scenario, such as methylation of transcription factor binding sites affecting RNA expression. However, this latter observation is not entirely new, as you point out yourself, and the statistical significance is difficult to judge without a genome-wide comparison. There is potential methylation of one PHR1 binding site, but what about all the other PHR1 binding sites in the genome, and what could be a plausible biological mechanism for such a specific effect? Finally, we appreciated the *Arabidopsis* data. We agreed that the differences in repeat content might explain the different effects seen in this species, but overall the *Arabidopsis* data do not help to better understand the phenomenon – with the added caveat that the experiments and analyses were not done exactly in the same way.

The agreement was that for *eLife*, causality of methylation and RNA expression changes would have to be established, using either chemical treatment or appropriate mutants. This would entail an entirely new set of experiments, and since it is *eLife's* policy to invite revision only when it is likely that the required experiments can be done in a short time frame, we are declining the study at this time. However, we think that this could in principle be appropriate for *eLife* with such added analyses, and we would reconsider such a substantially revised version of the work as a new submission.

*Reviewer #1*:

Secco and colleagues present one of the first single-nucleotide resolution studies on abiotic-stress-related methylation changes in plants. The experiment is well designed: rice plants are deprived of inorganic phosphate for up to 3 weeks, then are either kept under Pi deprivation or are re-supplied with Pi for another 1-31 days. The authors analyze gene expression by RNA-seq and DNA methylation by whole-genome bisulfite sequencing. The re-supply experiment is most noteworthy, as it shows very nicely a resetting of the transcriptional activity to control levels once Pi is available to the plant. The methylome, in contrast has a lag phase and only slowly returns back to the initial configuration, with some regions persistently maintaining their altered methylation pattern.

This study addresses the important question whether methylation can persistently change upon an abiotic stress signal and induce effects that would eventually allow the plant to adapt to a changing environment. Despite the quality of the data and the solid experimental setup, I have several major concerns:

1) I have had difficulties in assessing the general aim of the manuscript. On the one hand, the data presented are very descriptive, and most findings are not evaluated in detail. On the other hand, the data is presented in a way that suggests a regulatory or causal link between DNA methylation and transcription, although data support for this is only marginal or at least not verified. On several occasions the change in DNA methylation is described as being linked to the change in expression, as for the SPX2 locus, which is even given as an example of a putatively more general mechanism. In the Discussion, the authors then again refute links between methylation and expression. This latter take is much more consistent with their data (but is not kept up throughout the manuscript). For example, both hypo- and hyper-methylation-associated genes are up-regulated. Along the same line, re-establishment of control transcription levels does not correlate with re-established methylation after Pi resupply. SPX2 expression drops back to control levels after re-supply of Pi while the methylation is persistently altered. The conclusion from these observations should then be that gene expression and DNA methylation in response to Pi starvation are most likely not linked.

2) Another major concern is that many hypotheses built on the results from the methylome analysis are not verified, either statistically or experimentally, even though in many instances this would have been possible. There are numerous instances of claims about significant findings that lack statistical validation (explained below in more detail).

In the Abstract it reads that the detected changes in mC are “likely involved in the expression of repeat elements proximal to highly induced genes”. Why don't the authors use their transcriptome data or locus-specific analyses to address this important point?

3) The authors build their DMR detection on differentially methylated positions, which they then cluster into regions. It has been shown that for coverage and statistical reasons, DMPs are much easier to detect in the CG context than in CHG and CHH contexts. The DMR detection could therefore be biased towards 5mCG-rich regions. The authors should provide information on the DMPs and the distribution of the sequence contexts, so that it becomes clear whether DMR detection is skewed towards a specific context and thus towards certain genetic elements.

More importantly: why are the thresholds for DMR detection in the root and in the shoot different from each other (FDR<0.01 vs. FDR<0.05)? Could the lower threshold in the shoot explain why more DMRs are found in this tissue, which is not the primary Pi sensing organ?

What is the reasoning behind applying yet another FDR threshold for DMR detection in *Arabidopsis* (0.2 in this case)? It leaves the impression that the authors switch between different cut-off values in order to get some reportable DMRs where there are none or very few to report. Also, are the genes associated with these DMRs among the DEGs? It would be less confusing if the authors simply stated that no DMRs were detectable in *Arabidopsis*, as they allude to in the last paragraph of the Discussion.

Moreover, why was the sequencing for rice done on single plants while for *Arabidopsis* 10 plants were pooled in each replicate? Could it be that the higher number of DMRs is also due to high stochastic inter-individual variation? I suggest the authors compare differential methylation between replicates in both species. In any way it would be interesting to see the degree of that variation and how it scales to the variation between treatments. It should also be mentioned in the main text that the experimental setups significantly differed in this respect.

4) Changes in the CHH context imply that there is an involvement of the smallRNA machinery and the RdDM pathway. This is discussed nowhere in the manuscript.

5) The model on methylation-mediated binding of PHR1 to the P1BS binding motif is interesting, but it is completely speculative. Neither does the data show that expression of these genes is linked to methylation, nor do the authors provide evidence for the enhanced binding of PHR1 to the motif in its unmethylated state.

6) Transgenerational inheritance is only assessed in the context of persistence of several DMRs in the progeny of the stressed plants. As control I suggest that the authors identify DMRs between the two generations and check whether these DMRs are related to PSI loci. This way one might get an insight into DMRs induced in the germ line of the stressed plants, which would be the only ones possibly forwarded to the following generation.

Overall, the authors have generated a very interesting dataset, but, in terms of analysis and conclusions, in my view their study in its current state does not meet the standards of a high-profile publication.

*Reviewer #2*:

The manuscript by Secco and co-authors presents data on the identification of transcriptional and DNA methylation changes induced by phosphate (Pi) starvation and their dynamics in rice and to a smaller extent also *Arabidopsis*. In rice, part of the Pi starvation up- or down-regulated genes is associated with local DNA methylation changes, including some key Pi metabolism genes. Interestingly, these changes persist even after Pi re-supply, and based on this the authors suggest their mitotic heritability. However, they are not carried into the next generation. In contrast, Pi starvation induced DNA methylation changes were much more modest in *Arabidopsis*, indicating species-specific regulatory responses.

To my knowledge, this is so far the most comprehensive study addressing stress-induced DNA methylation changes. It provides important data towards the frequently proposed, but rarely evidenced, role of DNA methylation and transposable elements in controlling gene transcription.

I consider this study as an obvious candidate for publication in *eLife* upon revision.

*Reviewer #3*:

This study addresses the role of DNA methylation in plant gene regulation - specifically its role in transcriptional changes during phosphate starvation in rice. The authors perform a deep analysis of transcripts and DNA methylation using sequencing, following controlled phosphate starvation. They identify a set of genes that show transcriptional up-regulation together with coincident DNA demethylation in regions close to the TSS. Many of these genes are known to play functional roles in the phosphate starvation response. The authors demonstrate that many of these demethylated regions correspond to MITE transposons that have inserted close to the gene.

This is a thorough and detailed analysis, but it is essentially descriptive. It is unclear the extent to which the methylation changes are cause or effect of gene transcriptional induction. A number of mechanisms are possible, for example transcription may promote DNA demethylation via DNA glycosylases and/or inhibit DNA methylation maintenance. Mutations are available in rice DNA methylation factors, e.g. MET1 PMID 25002488 and DCL3 PMID 24554078, that may facilitate mechanistic experiments to test these ideas.

Transcriptional induction via transcription factors may be the most important factor here, which directly leads to demethylation of proximal TEs. In this scenario the methylation may not play an important role in control of gene expression. Removal of the methylation via mutation would allow expression of the target genes and associated TEs to be monitored to test the models proposed by the authors. It should also be possible to correlate Pol-II ChIP localisation with demethylation, in addition to other 5'-associated chromatin marks, e.g. H2A.Z, H3K4 methylation, nucleosome free regions. Further information is required to address how this methylation is removed and the extent to which it is really exerting an influence on the regulation of the genes of interest.

---

## [Author Response]

*We would ask you for some minor tweaks. First, it would be good if the title better reflected the new data, and that you refer to the temporal order of transcription and methylation changes*.

We have changed the title to: “Stress induced gene expression drives transient DNA methylation changes at adjacent repetitive elements”.

*Second, the reviewers were wondering whether there are phosphate-starvation-induced DMRs between* DCL3a *control and* DCL3a *stressed plants. The question that could be addressed with this is whether* DCL3a *is required for the stability of other regions in the genome that might not be the same as the stress-induced ones in the wild type*.

We have performed a new analysis accordingly. We have added a new figure (Figure 5—figure supplement 2) and describe the new data in the Results section (subsection “Stress-induced DMRs are mainly independent of the canonical RdDM pathway”). Briefly, it appears that while *DCL3a* could potentially mediate DNA methylation state of some genomic regions in response to Pi starvation, none of these were associated with changes in nearby gene expression.

*Finally, there are some minor changes, as listed below*.

Abstract

*Abstract. I find the word* “*proximal*” *a bit confusing. I would propose* “*nearby*” *or* “*close*”.

This has been changed throughout the manuscript.

*In the first paragraph of the subsection “Pi starvation induces widespread changes in DNA methylation, enriched at key regulators of Pi homeostasis”. I could not find how many ready had to cover a position in order to be considered as covered. This should be specified*.

We have modified a sentence in the main text to specify this as follows: “77-87% of cytosines covered by at least one read, 82-88% of genome”.

*The statement “the root PSI DMRs had an average size of 205 bp with a mean of 16 differentially methylated cytosines per DMR, and were preferentially localized within the first kilobase upstream (20%) and downstream (20%) of the proximal gene (*Figure 2*,*
Figure 2—figure supplement 2*)” and*
Figure 2
*may be potentially wrong. There are many more short intergenic regions than the long ones, which typically leads to the pattern described by the authors. In order to perform this analysis in an unbiased way, the authors should calculate this as percentage within individual intergenic regions*.

We have modified the analysis to take this into account, normalising the number of DMRs in each bin by the number of regions present in the corresponding bin. To do so, the position of each DMR was calculated with respect to the nearest gene. DMRs were then categorized in bins (within gene body, 0-1kb, 1-2kb, 2-4kb, 4-6kb, and >6 kb from the TSS or TES), and the number of DMRs in each bin was normalised by the total number of regions present in that bin category in the genome. Figure 2 and the main text have been updated as follows: “The root PSI DMRs… were preferentially localized within the first two kilobases upstream (40%) and first kilobase downstream (15%) of the nearby gene”.

*Discussion: “the differences in 439 DNA methylation changes observed in rice and* Arabidopsis *in response to Pi starvation could primarily be due to the differential frequency of TEs in the genome of each species, suggesting that* Arabidopsis *may not be the best model in which to study the role of DNA methylation in controlling TE activity in plants”. I think the large number of very successful genetic screens with results published in the top journals disproves this statement*.

This sentence has been removed from the revised manuscript.

[Editors’ note: the author responses to the previous round of peer review follow.]

*[…] The agreement was that for* eLife*, causality of methylation and RNA expression changes would have to be established, using either chemical treatment or appropriate mutants. This would entail an entirely new set of experiments, and since it is* eLife's *policy to invite revision only when it is likely that the required experiments can be done in a short time frame, we are declining the study at this time. However, we think that this could in principle be appropriate for* eLife *with such added analyses, and we would reconsider such a substantially revised version of the work as a new submission*.

In this substantially revised manuscript, we have addressed the issues raised by the reviewers about our initial submission, including the main point regarding establishing the causality of changes in DNA methylation and transcript abundance. We believe this is important given that there is a very frequent assumption that changes in DNA methylation drive changes in transcription. Accordingly, in this revised manuscript we have:

1) Added an earlier time point to our extensive phosphate starvation time course experiment, enabling us to establish the temporal hierarchy between changes in DNA methylation and transcript abundance. With this new data we clearly show that the changes in DNA methylation occur after the changes in transcript abundance. Thus, the observed DNA methylation changes are likely a consequence of increased transcription of the proximal stress responsive genes. Therefore, this study provides insights into a cellular activity that potentially repress the activity of specific transposable elements in genomic regions that must be transcriptionally activated to respond to environmental perturbation.

2) Performed a new phosphate starvation experiment, including the suggested *DCL3a* RNAi line, which is impaired in 24nt siRNA biogenesis. As suggested by reviewer 3, this new experiment sheds light on the mechanism involved in regulating these stress-induced changes in DNA methylation. Overall, we could show that while reduced expression of *DCL3a* had severe effects on the epigenome, altering DNA methylation at >9,000 regions, very few effects were observed in the stress induced DMRs, with hypermethylated DMRs being almost exclusively *DCL3a*-independent. This indicates that the stress induced DNA hypermethylation generally does not involve the canonical RdDM pathway.

We believe that this study now goes well beyond previous investigations into the dynamic modulation of DNA methylation in response to stress, establishes the causality between changes in transcript abundance and changes in DNA methylation, and shows that this process is mainly *DCL3a*-independent.

Reviewer #1:

*Secco and colleagues present one of the first single-nucleotide resolution studies on abiotic-stress-related methylation changes in plants. The experiment is well designed: rice plants are deprived of inorganic phosphate for up to 3 weeks, then are either kept under Pi deprivation or are re-supplied with Pi for another 1-31 days. The authors analyze gene expression by RNA-seq and DNA methylation by whole-genome bisulfite sequencing. The re-supply experiment is most noteworthy, as it shows very nicely a resetting of the transcriptional activity to control levels once Pi is available to the plant. The methylome, in contrast has a lag phase and only slowly returns back to the initial configuration, with some regions persistently maintaining their altered methylation pattern*.

*This study addresses the important question whether methylation can persistently change upon an abiotic stress signal and induce effects that would eventually allow the plant to adapt to a changing environment. Despite the quality of the data and the solid experimental setup, I have several major concerns*:

*1) I have had difficulties in assessing the general aim of the manuscript. On the one hand, the data presented are very descriptive, and most findings are not evaluated in detail. On the other hand, the data is presented in a way that suggests a regulatory or causal link between DNA methylation and transcription, although data support for this is only marginal or at least not verified. On several occasions the change in DNA methylation is described as being linked to the change in expression, as for the SPX2 locus, which is even given as an example of a putatively more general mechanism. In the Discussion, the authors then again refute links between methylation and expression. This latter take is much more consistent with their data (but is not kept up throughout the manuscript). For example, both hypo- and hyper-methylation-associated genes are up-regulated. Along the same line, re-establishment of control transcription levels does not correlate with re-established methylation after Pi resupply. SPX2 expression drops back to control levels after re-supply of Pi while the methylation is persistently altered. The conclusion from these observations should then be that gene expression and DNA methylation in response to Pi starvation are most likely not linked*.

We acknowledge that the message of the previous version could have been clearer, and we have endeavoured to clarify the message in the revised manuscript. In this revised version, we have included an additional and earlier time point (3 days + or - Pi) in order to decipher the temporal hierarchy between changes in DNA methylation and transcript abundance. The new data shows that changes in DNA methylation occur after changes in proximal gene transcription (Figure 3), thus indicating that changes in DNA methylation are likely a consequence of nearby induction of stress-responsive gene transcription. Overall, the revised manuscript clearly establishes the temporal relationship between differential transcription and DNA methylation.

*2) Another major concern is that many hypotheses built on the results from the methylome analysis are not verified, either statistically or experimentally, even though in many instances this would have been possible. There are numerous instances of claims about significant findings that lack statistical validation (explained below in more detail)*.

*In the abstract it reads that the detected changes in mC are* “*likely involved in the expression of repeat elements proximal to highly induced genes*”*. Why don't the authors use their transcriptome data or locus-specific analyses to address this important point*?

In the revised version of the manuscript, all claims about significant findings now include appropriate statistical validation. Due to extensive changes in the manuscript/Abstract, the Abstract sentence cited by Reviewer 1 is no longer present in the current version of the manuscript. Nonetheless, the following sentence has been added in the manuscript: “in addition, RNA-seq analysis failed to reveal any significantly differentially expressed TEs (Cuffdiff, FDR < 0.05) that overlapped with PSI DMRs, between Pi sufficient and deficient conditions (7 and 21 d), with more than 80% of them not being expressed”.

*3) The authors build their DMR detection on differentially methylated positions, which they then cluster into regions. It has been shown that for coverage and statistical reasons, DMPs are much easier to detect in the CG context than in CHG and CHH contexts. The DMR detection could therefore be biased towards 5mCG-rich regions. The authors should provide information on the DMPs and the distribution of the sequence contexts, so that it becomes clear whether DMR detection is skewed towards a specific context and thus towards certain genetic elements*.

We do not observe a bias towards DMR detection in mCG-rich regions. As shown in the manuscript Figure 2, most PSI DMRs occur in the CHH context, thus indicating that the DMR detection is not biased towards the CG context.

We agree that the identification of DMPs can be biased towards the CG context (Figure 8). However, comparison of the proportion of significantly differentially methylated cytosines located within the identified DMRs did not reveal an enrichment of DMPs in the CG context, suggesting that the identification of DMRs is unlikely to be biased towards the CG context.

Author response image 1.**DOI:**
http://dx.doi.org/10.7554/eLife.09343.036

*More importantly: why are the thresholds for DMR detection in the root and in the shoot different from each other (FDR<0.01 vs. FDR<0.05)? Could the lower threshold in the shoot explain why more DMRs are found in this tissue, which is not the primary Pi sensing organ*?

We agree with this comment. In the revised manuscript, all statistical analyses have been performed using a FDR threshold of 0.05, when the experiment was done using 3 biological replicates (n=3). However, due to the high number of replicates (n=9) used to identify root phosphate starvation induced (PSI) DMRs, a more stringent threshold of 0.01 was used, to detect a robust core set of PSI DMRs for further in depth analysis.

We agree that the higher FDR used for the shoot (FDR 0.05) is likely the cause of the increased DMRs identified in the shoots compared to the roots. However, we believe that the message of this comparison is clear: both roots and shoots have similar patterns of DNA methylation after phosphate starvation. We have modified the manuscript with respect to the reviewers comment, and clearly explained why different FDRs are being used.

*What is the reasoning behind applying yet another FDR threshold for DMR detection in* Arabidopsis *(0.2 in this case)? It leaves the impression that the authors switch between different cut-off values in order to get some reportable DMRs where there are none or very few to report. Also, are the genes associated with these DMRs among the DEGs? It would be less confusing if the authors simply stated that no DMRs were detectable in* Arabidopsis*, as they allude to in the last paragraph of the Discussion*.

We agree, and this has been clarified in the manuscript. In the revised version, we first used a FDR threshold of 0.05, showing that no DMRs could be identified using similar criteria as the rice DMR analysis. In addition, in order to assess whether any DMRs could be identified upon Pi starvation, a less stringent FDR threshold of 0.2 was used, resulting in the identification of only two PSI DMRs, proximal to phosphate-starvation induced genes.

*Moreover, why was the sequencing for rice done on single plants while for* Arabidopsis *10 plants were pooled in each replicate? Could it be that the higher number of DMRs is also due to high stochastic inter-individual variation? I suggest the authors compare differential methylation between replicates in both species. In any way it would be interesting to see the degree of that variation and how it scales to the variation between treatments. It should also be mentioned in the main text that the experimental setups significantly differed in this respect*.

While we agree with the reviewer’s comments, *Arabidopsis* roots are much smaller than rice roots. Consequently, the quantity of DNA recovered from individual roots (especially after 10 d of –Pi) was insufficient to perform whole genome bisulfite sequencing on single roots of a single *Arabidopsis* plant. The higher number of DMRs observed in rice is unlikely due to higher stochastic inter-individual variation since the main criteria for DMR identification was requiring 7 of the 9 replicates to show similar changes in DNA methylation, as mentioned in the methods. Comparison of the variation of the replicates in rice can be seen in Figure 2—figure supplement 1.

*4) Changes in the CHH context imply that there is an involvement of the smallRNA machinery and the RdDM pathway. This is discussed nowhere in the manuscript*.

We agree, and this has been revised in the current version of the manuscript. Indeed, we have now added substantial new experiments, repeating the phosphate starvation and DNA methylome/transcriptome experiment in a rice *DCL3a* RNAi line in order to decipher the involvement of the canonical RdDM pathway in Pi starvation induced DMRs. This demonstrated that the majority of the Pi starvation induced DMRs are *DCL3a*-independent.

*5) The model on methylation-mediated binding of PHR1 to the P1BS binding motif is interesting, but it is completely speculative. Neither does the data show that expression of these genes is linked to methylation, nor do the authors provide evidence for the enhanced binding of PHR1 to the motif in its unmethylated state*.

We agree, and this has been removed in the revised manuscript.

*6) Transgenerational inheritance is only assessed in the context of persistence of several DMRs in the progeny of the stressed plants. As control I suggest that the authors identify DMRs between the two generations and check whether these DMRs are related to PSI loci. This way one might get an insight into DMRs induced in the germ line of the stressed plants, which would be the only ones possibly forwarded to the following generation*.

In the current version of the manuscript, we assess the persistence of altered DNA methylation in all the Pi starvation induced DMRs, and clearly show that none of the PSI DMRs identified in the first generation are transmitted to the second generation.

In addition, as suggested, we investigated whether DMRs could be identified between the two generations, and if so whether they were related to any PSI loci. In order to identify DMRs between the two generations and exclude any potential DMRs that might arise from comparing different developmental stages, we compared the 3d +Pi time-point of the first generation of plants (15 day old plants: germinated for 2 weeks in +Pi, followed by 3 d of +Pi) to 10 d old seedlings from the parental plants that were continuously grown in + or –Pi conditions. Overall, only 21 and 24 DMRs (FDR < 0.05) could be identified between the first generation (3d +Pi) plants versus the progeny of Pi stressed parent and progeny of non-stressed plants, respectively. Assignment of these DMRs to their proximal gene showed that the majority of these DMRs were located near transposons and genes encoding ‘expressed proteins’. Importantly, none of these proximal genes showed a significant difference in transcript abundance in response to Pi stress (Cufflinks, FDR <0.05).

In conclusion, the ‘resetting’ of many root PSI DMRs over time after stress cessation, the very limited number of PSI DMRs identified in the panicles, the absence of persistent PSI DMRs in the progeny of stressed plants, and the analysis described above indicate that it is very unlikely that these stress induced DMRs are transmitted to the following generation.

Reviewer #3:

*[…] This study addresses the role of DNA methylation in plant gene regulation - specifically its role in transcriptional changes during phosphate starvation in rice. The authors perform a deep analysis of transcripts and DNA methylation using sequencing, following controlled phosphate starvation. They identify a set of genes that show transcriptional up-regulation together with coincident DNA demethylation in regions close to the TSS. Many of these genes are known to play functional roles in the phosphate starvation response. The authors demonstrate that many of these demethylated regions correspond to MITE transposons that have inserted close to the gene*.

*This is a thorough and detailed analysis, but it is essentially descriptive. It is unclear the extent to which the methylation changes are cause or effect of gene transcriptional induction. A number of mechanisms are possible, for example transcription may promote DNA demethylation via DNA glycosylases and/or inhibit DNA methylation maintenance. Mutations are available in rice DNA methylation factors, e.g. MET1 PMID 25002488 and* DCL3 *PMID 24554078, that may facilitate mechanistic experiments to test these ideas*.

We agree with this point, and as suggested by Reviewer 3 we performed extensive additional experiments to shed light on the mechanism and role of phosphate starvation induced (PSI) differential DNA methylation.

Firstly, as mentioned above (see Reviewer 1 section), we improved the already extensive time course experiment, allowing us to decipher the temporal relationship between changes in DNA methylation and transcript abundance. Indeed, we could show that changes in DNA methylation occurred after changes in nearby transcript abundance, thus suggesting that increased transcription of stress responsive genes could lead to increased methylation of nearby TEs.

Secondly, to gain further insights into the mechanism involved in the PSI differential DNA methylation, as suggested by Reviewer 3, we performed a new experiment utilizing the suggested rice *DCL3a* RNAi line. We grew WT and a *DCL3a* RNAi line (PMID 24554078), which is impaired in 24nt siRNA biogenesis, under + or –Pi for 21 days, before performing MethylC-seq and RNA-seq. While reduced expression of *DCL3a* had major effects on the epigenome, altering DNA methylation at >9000 regions in the genome, very few effects were observed in the phosphate starvation induced DMRs, with hypermethylated DMRs being almost exclusively *DCL3a*-independent, indicating that this process likely does not involve the canonical RdDM pathway.

The suggested *met1-2* mutant was not included in this new experiment due to the extremely deleterious effects of the mutation on plant growth and morphology: “…growth of the seedlings was significantly retarded, and they could only grow to ca. 3 cm in height with sporadic, deformed tiny roots, and then growth of the seedlings was totally halted and underwent necrotic death within 2 wk” (Hu et al., 2014 PMID 25002488).

*Transcriptional induction via transcription factors may be the most important factor here, which directly leads to demethylation of proximal TEs. In this scenario the methylation may not play an important role in control of gene expression. Removal of the methylation via mutation would allow expression of the target genes and associated TEs to be monitored to test the models proposed by the authors. It should also be possible to correlate Pol-II ChIP localisation with demethylation, in addition to other 5'-associated chromatin marks, e.g. H2A.Z, H3K4 methylation, nucleosome free regions. Further information is required to address how this methylation is removed and the extent to which it is really exerting an influence on the regulation of the genes of interest*.

While we agree with reviewer 3, we believe the model on methylation-mediated binding of PHR1 to the P1BS binding motif is too speculative at this stage and has thus been removed from the revised manuscript.

However, to shed light on the mechanism involved in the regulation Pi starvation-induced DMRs, as suggested by reviewer 3, we used a *dcl3a* RNAi line (PMID 24554078). Briefly, we now show that the majority of phosphate starvation induced DMRs are not dependent upon *DCL3a* and thus do not involve the canonical RdDM pathway. These findings have important implications for interpreting the capacity of environmentally induced epialleles to influence genic transcription, and their stability through plant growth and reproduction.